# Endogenous aldehyde accumulation generates genotoxicity and exhaled biomarkers in esophageal adenocarcinoma

Stefan Antonowicz [1], Zsolt Bodai[2], Tom Wiggins[1], Sheraz R. Markar[1], Piers R. Boshier[1], Yan Mei Goh [1], Mina E. Adam[1], Haonan Lu[1], Hiromi Kudo[1,3], Francesca Rosini[3], Robert Goldin [3], Daniela Moralli [4], Catherine M. Green[4], Chris J. Peters[1], Nagy Habib[1], Hani Gabra [1], Rebecca C. Fitzgerald [5], Zoltan Takats[2] & George B. Hanna [1✉]

Volatile aldehydes are enriched in esophageal adenocarcinoma (EAC) patients' breath and could improve early diagnosis, however the mechanisms of their production are unknown. Here, we show that weak aldehyde detoxification characterizes EAC, which is sufficient to cause endogenous aldehyde accumulation in vitro. Two aldehyde groups are significantly enriched in EAC biopsies and adjacent tissue: (i) short-chain alkanals, and (ii) medium-chain alkanals, including decanal. The short-chain alkanals form DNA-adducts, which demonstrates genotoxicity and confirms inadequate detoxification. Metformin, a putative aldehyde scavenger, reduces this toxicity. Tissue and breath concentrations of the medium-chain alkanal decanal are correlated, and increased decanal is linked to reduced *ALDH3A2* expression, *TP53* deletion, and adverse clinical features. Thus, we present a model for increased exhaled aldehydes based on endogenous accumulation from reduced detoxification, which also causes therapeutically actionable genotoxicity. These results support EAC early diagnosis trials using exhaled aldehyde analysis.

[1] Department of Surgery and Cancer, Imperial College London, London, UK. [2] Department of Metabolism, Digestion and Reproduction, Imperial College London, London, UK. [3] Centre for Pathology, Imperial College London, London, UK. [4] Chromosome Dynamics Core, Wellcome Centre for Human Genetics, University of Oxford, Oxford, UK. [5] MRC Cancer Unit, Hutchison/MRC Research Centre, University of Cambridge, Cambridge, UK. ✉email: g.hanna@imperial.ac.uk

Alerting symptoms of esophageal adenocarcinoma (EAC) typically indicate advanced and incurable disease, whereas features of early stage disease, such as heartburn, are non-specific[1,2]. Thus, despite the availability of safe and effective treatments for early EAC, prognosis remains poor with a 5-year survival of 14%[3]. The gold standard diagnostic test is endoscopy, which is invasive, uncomfortable, and expensive. Additionally, this cannot be offered to every patient with non-specific symptoms, as these are extremely common. New diagnostic adjuncts are needed to triage patients with non-specific symptoms to receive an endoscopy, especially as the EAC incidence in Western countries has risen sharply in recent decades and is not projected to fall[4].

Exhaled aldehydes have been consistently demonstrated to be enriched in EAC patients[5,6], and could form the basis of a non-invasive, primary care-based, triage test for patients with non-specific upper gastrointestinal symptoms. However, the mechanisms underlying these biomarkers' production remain unknown. Molecular studies have identified that endogenous aldehydes are potent mediators of malignant transformation[7,8]. These carbonyls can react with base-pairing amines in DNA at ambient conditions, and formaldehyde and acetaldehyde among others are designated carcinogens[9]. For esophageal squamous cancers, inactivating variants in *ALDH2* convey predisposing risk[10]. For EAC and the precursor esophageal intestinal metaplasia (Barrett's esophagus), glutathione defenses—an auxiliary mechanism of aldehyde detoxification—may be depleted[11,12]. Together, this suggests the esophagus is exposed to aldehyde stress, and that glandular esophageal cells may be vulnerable to this stress. However, esophageal aldehyde biochemistry has not been described.

In this study, we investigate whether exhaled EAC biomarkers could be explained by corresponding changes in the EAC microenvironment, and aim to identify the molecular mechanisms underpinning these changes. We show that short-chain genotoxic alkanals and also medium-chain alkanals are specifically enriched in EAC and associated tissues, on a genetic background of reduced aldehyde detoxification. This metabolic deregulation generates genotoxicity and correlates with exhaled biomarkers, and thus has implications for ongoing research in early EAC detection.

## Results

**Reduced aldehyde detoxification characterizes glandular tissues in the esophagus.** We first asked whether a cell-autonomous mechanism could contribute to exhaled biomarkers, so we began by searching for thematic differences in metabolic gene expression between EAC and relevant control tissue in archived transcriptomic datasets. Compared to EAC, squamous mucosa for healthy controls (SqN) was defined by a strong aldehyde oxidation theme (Fig. 1a, b and Supplementary Data 3), suggesting reduced aldehyde defense in EAC tissue. Ingenuity Pathway Analysis identified an analogous leading phenotype (Supplementary Fig. 1a). The EAC precursor Barrett's metaplasia also had differences in redox gene expression (Supplementary Data 3), in keeping with a previous report of disengaged glutathione metabolism[11]. A meta-analysis of the public datasets at a single gene level identified *ALDH1A3, -3A1, -3A2, -4A1*, and *-9A1* as consistently under-expressed in EAC compared to normal esophageal squamous mucosa (Supplementary Fig. 1b, *ALDH3B2* is a pseudogene). We validated these observations experimentally using quantitative polymerase chain reaction (qPCR) on paired endoscopic biopsies from 67 patients (Fig. 1c). These data complement those from epidemiological and genomic studies which link deactivating *ALDH* variants to esophageal malignancies[10,13].

Importantly, these five aldehyde dehydrogenase isoenzymes show specificity to products of lipid peroxidation, a major source of endogenous aldehydes[14].

Further assessments suggested that *ALDH3A2* was expressed in BAR but not EAC, whereas other ALDH genes were under-expressed in both BAR and EAC (Supplementary Fig. 1b, c). To explore this, we used immunohistochemistry, as non-dysplastic Barrett's epithelium is a monolayer and prone to contamination in bulk tissue preparations (see Fig. 1d, e). *ALDH3A1* was strongly expressed in SqT but weakly in Barrett's tissue and EAC (Fig. 1d, f). In contrast, its 17p co-localizing homolog *ALDH3A2* was expressed in all SqT, >90% of Barrett's metaplasia and dysplasia, and 21% of EAC (Fig. 1e, f). Subgroup analysis by clinicopathologic features revealed associations of reduced *ALDH3A2* with advanced stage and poorly differentiated disease (Supplementary Data 4). Thus, low expression of *ALDH3A1* was common to all esophageal glandular epithelia, whereas loss of *ALDH3A2* was linked to malignant transformation and progression. These genes' products show particular affinity to medium and fatty alkanals[14], and both genes are not expressed in most EAC cases.

Next we tested whether squamous and EAC in vitro models shared these phenotypes. Good commercial esophageal squamous models are lacking[15], so we validated a new technique for culturing normal esophageal keratinocytes in vitro using the rho kinase inhibition/irradiated fibroblast co-culture method[16]. These cells strongly expressed squamous cytokeratins, p63, and E-cadherin, and were depleted for vimentin (Supplementary Fig. 1d). Compared to commercial EAC and Barrett's cell lines, keratinocyte cultures had robust expression of all tested *ALDH* genes, and lower concentrations of DNA double-strand break marker phosphorylated $\gamma$-H$_2$AX (Supplementary Fig. 1e).

Recent work suggests that ALDH-mediated detoxification is the primary means of preventing endogenous aldehyde accumulation[7]. To test whether the observed reduced *ALDH* expression could cause aldehyde accumulation in an EAC-relevant system, we blocked *ALDH* pharmacologically and genetically, and measured metabolic phenotypes. To do this, we developed a targeted ultra-performance liquid chromatography tandem mass spectrometry (UPLC-MS/MS) method to quantify 42 carbonyls (aldehydes and ketones) in tissue and liquid samples, updating the classical dinitrophenylhydrazine (DNPH) approach (see "Methods"). The method overcame the range in polarity, volatility and reactivity from light to fatty acyls, covered a broad range of biology-relevant carbonyls, and allowed unambiguous identification of DNPH-reactive isomers/isobars such as alkanals, ketones, and dialdehydes (e.g. acetone, propanal, glyoxal, formula mass = 58). There is metabolic redundancy between the 19 ALDH isoenzymes[14], so we initially chose diethylaminobenzaldehyde-mediated inhibition, as this inhibits nearly all of the isoenzymes[17]. There was dose-dependent enrichment of measured alkanals in OE33 cells, suggesting that ALDH inhibition is sufficient to overcome alternative detoxification mechanisms and cause endogenous aldehyde accumulation (see Supplementary Fig. 2a). Similar effects were observed under hypoxic conditions (see Supplementary Fig. 2b), although aldehyde concentrations subsequently reduce, implying metabolic adaptation. Combining hypoxia and ALDH inhibition led to sustained aldehyde increases (see Supplementary Fig. 2c). These phenotypes were replicated in a second EAC cell line (Supplementary Fig. 2d–f).

We then used RNA interference to study the contribution of each differentially expressed ALDH isoenzyme to endogenous aldehyde metabolism. Silencing individual ALDH isoenzymes in OE33 cells under normal conditions led to enrichment of acetaldehyde, nonanal, decanal, and some enals (see Supplementary Fig. 2g, h). These effects were more pronounced when

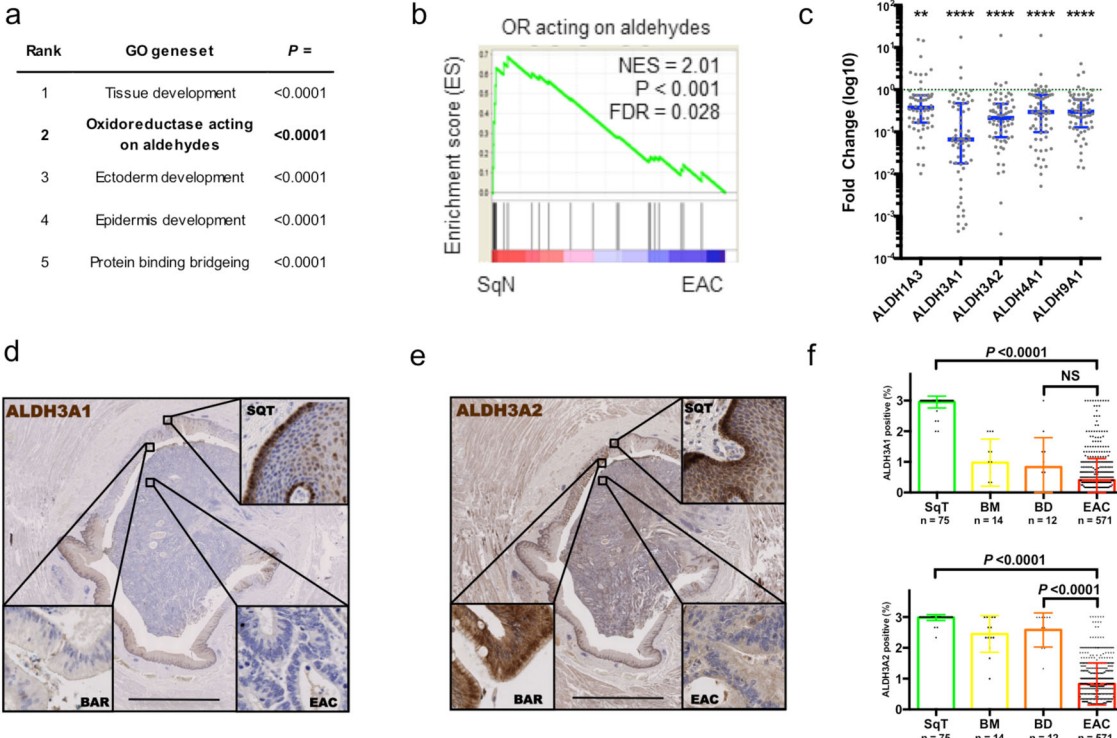

**Fig. 1 Loss of aldehyde detoxification is conspicuous in esophageal adenocarcinoma. a** Geneset enrichment analysis comparing squamous mucosa (SqN, $n = 19$) and esophageal adenocarcinoma (EAC, $n = 21$) samples in GSE26886. Five most significantly enriched Gene Ontology v4.0 genesets in squamous mucosa. $P$ values calculated by the permutation test in the GSEA analysis. **b** Enrichment plot of "oxidoreductase (OR) acting on aldehydes" geneset from **a**. **c** Relative expression of candidate ALDH isoenzymes in EAC endoscopic biopsies compared to paired SqT biopsies ($n = 67$), quantitative reverse transcriptase polymerase chain reaction (median and 95% confidence interval provided. $P$ values calculated with two-sided Wilcoxon matched-pairs signed rank test; ALDH1A3 $P = 0.0026$, all others $P < 0.0001$. **d**, **e** Representative immunostaining for ALDH3A1 (**d**) and ALDH3A2 (**e**) in an esophageal cross-section featuring an adenocarcinoma (bar indicates 5 mm). **f** Summary of ALDH3A1 and -3A2 immunostaining in cancer-adjacent squamous tissue (SqT, $n = 75$), Barrett's metaplasia (BM, $n = 14$; Barrett's dysplasia (BD, $n = 12$), and EAC ($n = 571$) drawn from nine UK hospitals. Mean ± SEM provided, Kruskall–Wallis test with Dunn's correction **$P < 0.01$, ****$P < 0.0001$. Source data are provided in the Source Data file.

multiple genes were silenced, and led to stabilization of the genotoxicity marker phosphorylated γ-H2AX (Supplementary Fig. 2g). Similar metabolic phenotypes were observed on silencing ALDH in CPD cells (Supplementary Fig. 2i). This confirms that reduced ALDH activity is sufficient to enrich endogenous aldehydes in glandular esophageal cell lines; however, loss of specific isoforms causes subtle effects compared to broader genetic interference or environmental influences.

**Genotoxic and medium-chain aldehydes are enriched in the adenocarcinoma-bearing esophagus.** These results encouraged us to measure EAC aldehyde concentrations in situ, using parallel endoscopic samples from the PCR study and the optimized UPLC-MS/MS method. Metabolic field effects have been described in the malignant esophagus[18], so we chose four test groups: adenocarcinoma (EAC), cancer-adjacent squamous epithelium (SqT), cancer-adjacent Barrett's metaplasia (BAR), and squamous epithelium from patients without endoscopic abnormality (SqN) (see Supplementary Data 5).

In unsupervised multivariate analysis, SqN separated from other tissue in the first principal component, which explained 47% of the data (Fig. 2a). EAC tissues were significantly enriched for 20 of 27 of the target aldehydes, compared to SqN (Supplementary Fig. 3). Correlating aldehyde concentrations in EAC samples revealed three correlated groups: short-chain, medium-chain and fatty alkanals (Fig. 2b). The genotoxic aldehydes formaldehyde, acetaldehyde, and 1-butenal were significantly increased in both SqT and EAC compared to SqN (Fig. 2c–e), suggesting aldehyde

stress exhibits a field effect in the transformed esophagus. EAC was also enriched for other genotoxic aldehydes, including glyoxal, malondialdehyde, and 4-hydroxy-2-nonenal (HNE) (Fig. 2f and Supplementary Fig. 3).

In contrast, medium-chain alkanals were markedly concentrated in EAC compared to SqT and SqN, suggesting cell-autonomous production (see Fig. 2g–j and Supplementary Fig. 3). Of particular interest was decanal (SqN vs EAC, relative difference 1.54, $P < 0.05$), a consistent EAC biomarker in exhaled breath studies[5,6]. There were no significant differences in fatty alkanal concentrations, despite these metabolites being tightly correlated. Thus, we selected the genotoxic and medium-chain groups for further focused study.

**Genotoxic aldehydes are present as DNA-adducts and are therapeutically targetable.** To establish whether the genotoxic aldehydes affect esophageal DNA health, we used a second UPLC-MS/MS method[19] to quantify aldehyde-DNA damage products in patient tissue. Two purine nucleosides were selected, which feature base-pairing surfaces covalently distorted by either acetaldehyde, or a naturally occurring[20] HNE derivative (CrodG and 1N^6-εdA, respectively, Fig. 3a). Compared to SqN and circulating leukocyte DNA, both SqT and EAC isolates were significantly enriched for both tested DNA damage products (Fig. 3b). These results complement the tissue-aldehyde series, verifying aldehyde accumulation in EAC and adjacent tissue. Given the base-pairing position of these adducts, it also proposes these aldehydes as potential esophageal mutagens.

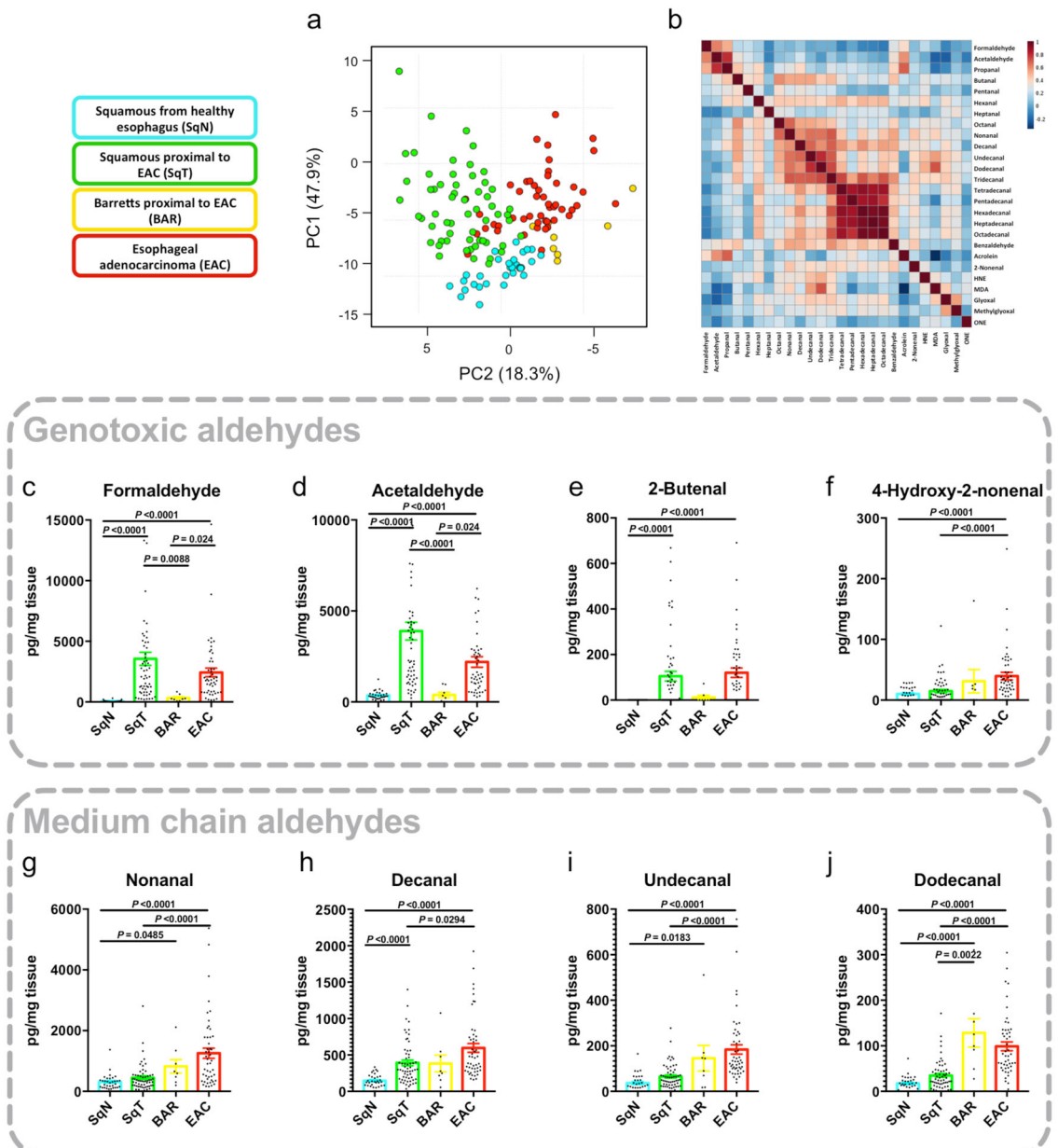

**Fig. 2 Biopsies of esophageal adenocarcinoma and adjacent squamous tissue samples are enriched for genotoxic and medium-chain aldehydes.**
**a** Principal component analysis (PCA) of esophageal tissue aldehyde profiles measured by UPLC-MS/MS. Squamous tissue from healthy patients (SqN, $n = 32$ independent patients), esophageal adenocarcinoma (EAC)-adjacent squamous tissue (SqT, $n = 59$), EAC-adjacent Barrett's (BAR, $n = 8$), and EAC ($n = 59$) were compared. **b** Correlation matrix of aldehydes in EAC tissue samples in **a**. Scale indicates Pearson's $r$. **c–l** Absolute concentrations of the indicated aldehydes in the tested tissues. Mean ± SEM provided, Kruskall–Wallis test with Dunn's correction, $*P < 0.05$, $**P < 0.01$, $***P < 0.001$, $****P < 0.0001$. Source data are provided in the Source Data file.

The availability of medications[21] which scavenge aldehydes or promote their detoxification prompted us to explore whether this biology could be targeted therapeutically. We screened a panel of agents using a 48 h viability assay in FLO1 cells, and found that pretreatment with metformin offered a significant survival advantage following exposure to acetaldehyde (relative absorbance 1.27, 95% CI 1.23–1.30, $P < 0.0001$, Supplementary Fig. 4a). Metformin is a biguanide with several nucleophilic moieties capable of addition reactions with reactive carbonyls[22], and has been shown to reduce aldehyde-mediated genotoxicity in hematological malignancies[23]. A re-analysis of a previously published breath dataset[6] revealed that diabetics with EAC who were taking metformin had several significantly reduced alkanals

compared to those not taking metformin (Supplementary Fig. 4b). Initial experiments revealed that 48-h pretreatment with pharmacologically relevant concentrations of metformin could prevent stabilization of phosphorylated γ-H2AX in a panel of esophageal cell lines, following a 5-h exposure to pathologically relevant concentrations of formaldehyde or acetaldehyde (Fig. 3c, d). Further functional assessments of the DNA damage response indicated that metformin caused a dose-dependent suppression of dsDNA-mediated ATM/Chk2 signaling in EAC and Barrett's cell models (Fig. 3e, f), improved viability at a range of acetaldehyde concentrations (Supplementary Fig. 4c, d), and increased intracellular stores of reduced glutathione (Supplementary Fig. 4e, f). In summary, there is evidence of

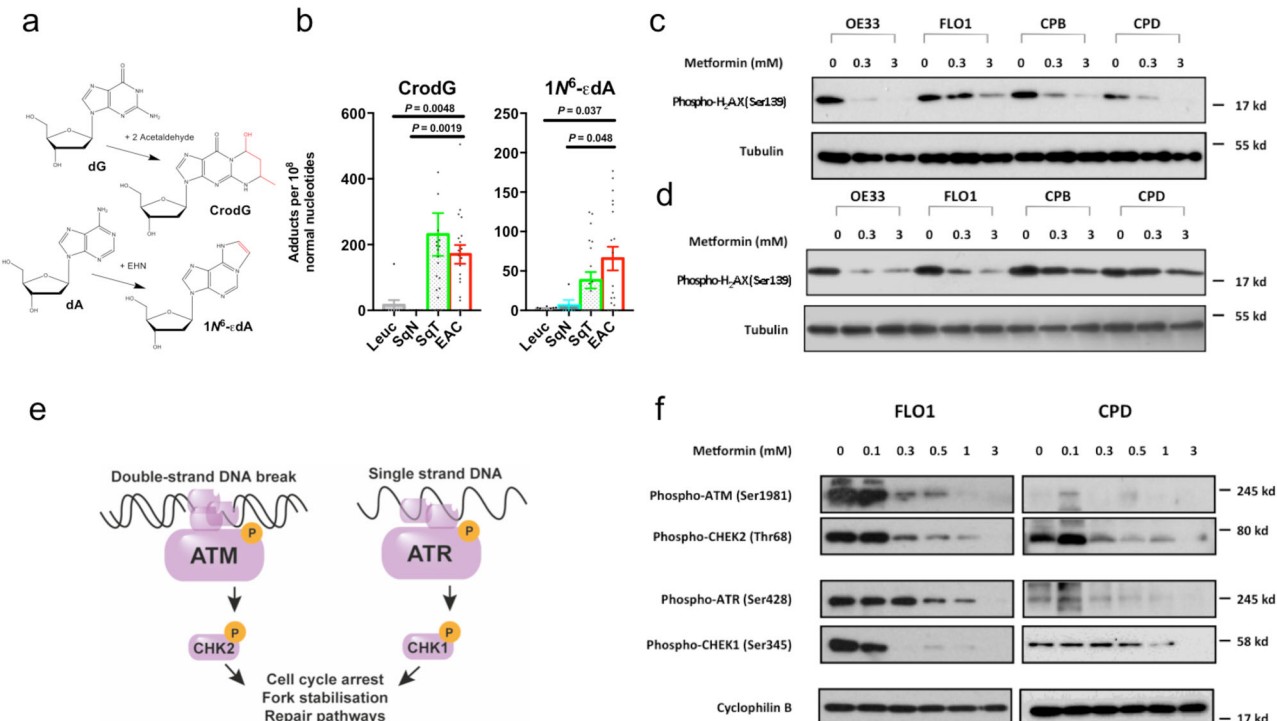

**Fig. 3 Aldehyde DNA damage is present in esophageal adenocarcinoma, which can be reduced in vitro by metformin. a** Schematic of deoxyadenosine and deoxyguanosine adduction by acetaldehyde (C2:0) and epoxy-hydroxynonanal (EHN), respectively, to form CrodG and $1N^6$-εdA. **b** CrodG and $1N^6$-εdA concentrations in DNA extracts from peripheral leukocytes ($n = 9$), squamous tissue from healthy patients SqN ($n = 5$), esophageal adenocarcinoma-adjacent squamous tissue (SqT, $n = 20$), and esophageal adenocarcinoma (EAC, $n = 19$). **c, d** Western blot analysis of the indicated cell lines pre-treated with metformin (MET) and exposed to 150 μM formaldehyde or 500 μM acetaldehyde for 5 h. **e** Schematic of the ATM and ATR-mediated DNA damage repair **f** Western blot analysis for DNA damage repair protein phosphorylation in the indicated cell lines pre-treated with metformin and exposed to 500 μM acetaldehyde for 5 h. Mean ± SEM provided for analytical triplicates of biological duplicates or triplicates, two-tailed Mann–Whitney $U$ test or ANOVA, *$P < 0.05$, **$P < 0.01$, ***$P < 0.001$, ****$P < 0.0001$. All blots are representative of three independent experiments with similar results. Source data are provided in the Source Data file.

aldehyde genotoxicity in EAC, which can be modulated by metformin in vitro.

**Decanal links tissue metabolism to exhaled diagnostic biomarkers.** Next we compared the EAC tissue aldehyde profile to the breath aldehyde profile for similarities. We first used our previously published[6] biomarker search methodology to identify metabolic features which independently contributed to a multi-variable diagnostic model for EAC, based on tissue aldehydes. A five-metabolite model had an area-under-the-curve of 0.983, indicating excellent discriminatory performance (Fig. 4a). Decanal was a feature of this model.

Given that the validated diagnostic breath model[6] for EAC contains butanal and decanal, we then checked whether the tissue and breath concentrations were correlated. To bridge the gap between the tissue and breath metabolic pools, we also tested whether the concentration of these aldehydes significantly differed in the headspace above EAC tissue, compared to normal healthy mucosa (independent cohort, EAC $n = 25$, SqN $= 25$), using our validated proton transfer reaction time-of-flight mass spectrometry method (see Supplementary Fig. 5). Tissue butanal concentration was not discriminatory for EAC, did not correlate to exhaled butanal, and was not significantly different in tissue headspace (Fig. 4b–d). In contrast, tissue decanal concentrations were discriminatory for EAC, were correlated with exhaled decanal, and were significantly increased in EAC tissue headspace (Fig. 4e–g). These results reaffirm decanal as an EAC biomarker, and suggest that exhaled decanal, but not exhaled butanal, may originate within the

tumor. The tissue headspace analysis also identified enriched acetaldehyde, hexanal, nonanal, and undecanal from EAC samples (see Supplementary Fig. 5), which provides cross-platform validation with the UPLC-MS/MS findings (Fig. 2).

**Decanal is prognostic and linked to TP53.** Finally we wanted to understand the mechanism for intra-tumoral decanal production. *ALDH3A1* and *-3A2* have activity against medium- and fatty-alkanals[14], and silencing of either enriched nonanal and decanal concentrations in vitro (Supplementary Fig. 2h, i). Similarly, *ALDH3A1* and *-3A2* low EAC tumors were enriched for decanal and other medium-chain alkanals (Fig. 5a, b), and expression of those genes inversely correlated to exhaled decanal concentrations (Fig. 5c, d). Thus, in EAC in vivo and in vitro, reduced *ALDH3A1* and *-3A2* activity is associated with increased decanal.

In The Cancer Genome Atlas (TCGA) data, reduced *ALDH3A2* expression in EAC was associated with significantly poorer survival (Supplementary Fig. 6a), a pattern we verified in the OCCAMS immunophenotyping cohort ($n = 360$, log-rank $P = 0.0001$, Fig. 5e). *ALDH3A2* expression loss was also associated with transformation to invasive disease, and later T-stage (Fig. 1e–f and Supplementary Data 4). On multivariable regression, low *ALDH3A2* expression independently predicted death (odds ratio 1.64, 95% CI 1.13–2.39, $P = 0.01$, Supplementary Data 6). Similarly, analysis of a previous breath dataset[6] linked high exhaled decanal to later T-stage and significantly poorer survival (Fig. 5f and Supplementary Fig. 6b). Increased tissue decanal was also associated with poorer overall survival

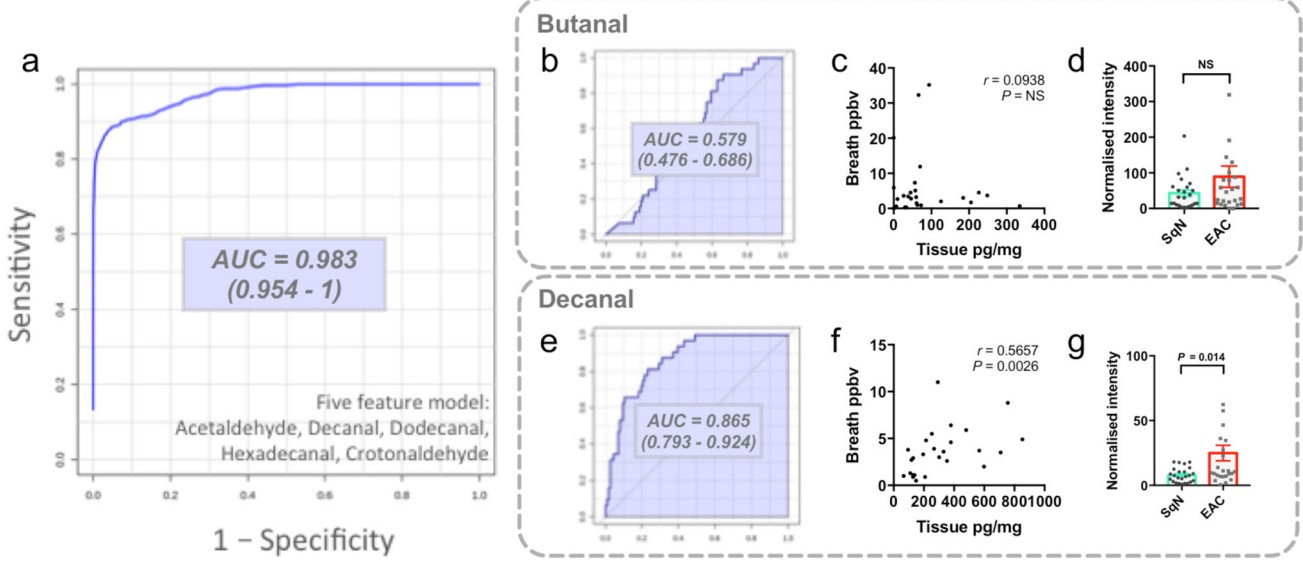

**Fig. 4 Tissue decanal is diagnostic and correlates to exhaled decanal. a** Receiver operating characteristic curve for EAC detection using a random forests multivariable model based on the indicated tissue aldehydes, measured by UPLC-MS/MS. **b–g** Comparison of tissue and exhaled butanal and decanal. **b, e** The receiver operating characteristic curve based on tissue butanal (**b**) or decanal (**e**) concentrations as a single biomarker (UPLC-MS/MS cohort). **c, f** Correlation of EAC tissue butanal (**c**) and decanal (**f**) concentration to the paired exhaled breath concentration (data from Markar et al.[6]). $P$ values calculated with a two-tailed Spearman test. **d, g** Gas-phase concentrations of butanal (**d**) and decanal (**g**) in the headspace of healthy squamous (SqN, $n = 24$) and esophageal adenocarcinoma (EAC, $n = 25$) samples measured by proton transfer reaction time-of-flight mass spectrometry. Mean ± SEM provided with two-tailed Mann–Whitney $U$ test. Source data are provided in the Source Data file.

(Fig. 5g). In summary, both reduced *ALDH3A2* and enriched decanal are associated with EAC progression and poorer survival.

Congenital *ALDH3A2* mutations cause Sjogren–Larsson syndrome (SLS), a neuro-cutaneous syndrome in which keratinocyte hyperplasia and fatty alkanal accumulation are features[21,24]. This strengthens the argument that *ALDH3A2* is metabolically non-redundant. Given that loss of *ALDH3A2* (and enriched decanal) was associated with EAC progression, whereas *ALDH3A1* is generally suppressed in all glandular esophageal cell types, we then focused on the regulation of *ALDH3A2* as the key mediator of decanal metabolism in EAC development. In TCGA data, the regulatory regions of *ALDH* genes were not commonly mutated or methylated (Supplementary Fig. 6c). However, in transcriptomic data, *ALDH3A2* expression was most highly correlated with telomeric neighbors (Supplementary Fig. 6d), and an analysis of *ALDH3A2* expression in The Cancer Cell Line Encyclopedia dataset showed a positive correlation with copy number (Supplementary Fig. 6e). Hypothesizing that *ALDH3A2* copy influences *ALDH3A2* expression in EAC, we first checked for focal or regional copy change at 17p11 by applying GISTIC[25] to TCGA single-nucleotide polymorphism array data. This suggested that 17p frequently undergoes whole-arm shallow deletion, affecting both the *ALDH3A1/2* and *TP53* loci (Fig. 5h). Chromosome 17p loss-of-heterozygosity is a common event in transformation to EAC and conveys poorer prognosis[26], and *TP53* is the only recurrently deactivated gene in EAC (>80% cases)[27,28]. Importantly, in two public datasets, comparing copy number to expression revealed a positive correlation for *ALDH3A2* but not *ALDH3A1* (Supplementary Fig. 5f–k).

To confirm this experimentally, we sought to compare *ALDH3A2* copy number with expression in the OCCAMS immunophenotyping cohort ($n = 360$), using four-color fluorescence in situ hybridization (FISH), confocal image acquisition in three dimensions, and automated image analysis (Fig. 5i). This uses cell morphology to restrict analysis, and thus minimizes signal contamination from undissectable normal populations

(e.g., fibroblasts, endothelia, immune cells). Tumors with reduced *ALDH3A2* immunostaining had a significantly reduced *ALDH3A2* copy ratio, although *ALDH3A1* copy number had no effect on *ALDH3A1* expression (Fig. 5j). We then tested the hypothesis that 17p arm deletion involves both *TP53* and *ALDH3A2* loci by comparing corresponding FISH signal ratios. These counts were proportionally related (Fig. 5k), in keeping with TCGA and ICGC genomic data, which supports the hypothesis that *ALDH3A2* copy loss and reduced expression is related *TP53* copy loss. Given that *TP53* inactivation occurs during progression to high-grade dysplasia or EAC[27], this could explain why *ALDH3A2* loss was linked to progression in the immunophenotyping experiments (Supplementary Datas 4 and 6), although SLS observational studies have not reported increased EAC incidence[21]. *ALDH3A1* expression was not affected by copy number aberration, presumably because glandular cells do not express this gene anyway (Fig. 1d, f and Supplementary Fig. 1b). In summary, reduced *ALDH3A2* expression is associated with 17p copy loss involving *TP53*, and inversely related to tissue and exhaled decanal. This effect was associated with poor clinical prognosis.

**Discussion**

In this study we investigated how intra-tumoral metabolism contributes to exhaled diagnostic biomarkers of EAC. We found suppressed aldehyde oxidation to be an EAC hallmark, consistent with observations of impaired glutathione-mediated redox defense in Barrett's and EAC[11], and epidemiological series which linked inactivating *ALDH* variants to esophageal malignancies[10,13]. In vitro, we show aldehyde oxidation loss to be sufficient to enrich endogenous aldehydes, in keeping with recent suggestions that detoxification is the primary means of controlling endogenous aldehyde accumulation[7]. We then demonstrated increased concentrations of genotoxic and medium-chain alkanals in EAC samples and adjacent tissues, which correspond to clinical series measuring exhaled alkanals in EAC[5,6]. Finally, we

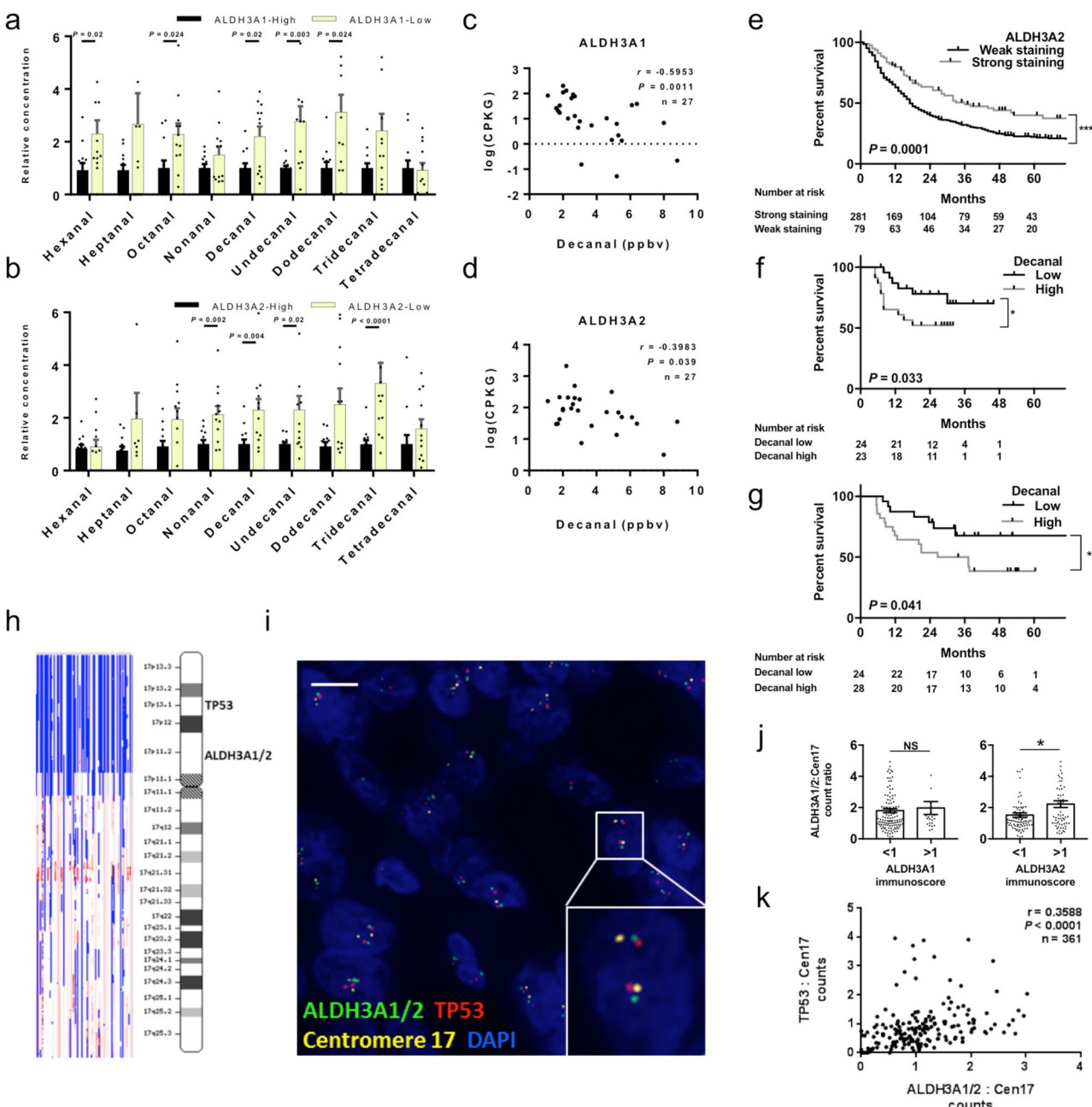

**Fig. 5 ALDH3A2 loss connects decanal to prognosis and TP53. a, b** Relative aldehyde concentrations in tissue samples of esophageal adenocarcinoma (UPLC-MS/MS cohort) divided into *ALDH3A1* (**a**) and *ALDH3A2* (**b**) high or low tumors (n = 24, two equal groups). **c, d** Correlations of *ALDH3A1* (**c**) and *ALDH3A2* (**d**) expression (CPKG, copies-per-thousand GAPDH values, n = 27) to paired exhaled decanal concentrations from Markar et al.[5]. **e** Overall survival in the multi-center immunohistochemistry cohort, stratified by *ALDH3A2* expression (immunoscore >1 used as cut-off), log-rank test. Patients surviving <3 months from surgery were excluded (total included n = 329). **f** Overall survival for patients with esophageal adenocarcinoma in the previously published exhaled breath cohort[5], dichotomized about the median (n = 47). Patients surviving <3 months were excluded. **g** Overall survival for patients with esophageal adenocarcinoma in the present UPLC-MS/MS tissue aldehyde dataset, dichotomized about the median (n = 52). Patients surviving <3 months were excluded. **h** GISTIC analysis of single-nucleotide polymorphism array data from The Cancer Genome Atlas (n = 89). Blue indicates shallow deletion red indicates copy gain. **i** Representative fluorescent in situ hybridization (FISH) image of EAC, bar indicates 25 μm (false colors used). **j** *ALDH3A1/2* FISH signal counts compared to *ALDH3A1* and *ALDH3A2* expression (cut-off immunoscore >1 used, n = 145, P = 0.0135 for *ALDH3A2*). **k** Correlation of *TP53* FISH signals to *ALDH3A1/2* FISH signals. One dot = one patient (n = 134). Pearson's test. Mean ± SEM provided, two-tailed Mann–Whitney U-test used unless otherwise stated, *P < 0.05, **P < 0.01, ***P < 0.001. Source data are provided in the Source Data file.

show that certain aldehydes contribute to genotoxicity in esophageal tissues, whereas others are linked to key genetic events in progression, and convey prognostic information. These data provide a model in which reduced detoxification causes the accumulation of volatile endogenous aldehydes, and thus a mechanistic basis for a diagnostic breath test for EAC based on aldehyde biochemistry. This is of critical importance, as the incidence of EAC has increased sharply in recent decades, survival remains poor, and diagnosis at treatable disease stages is fundamental to improving prognosis.

Our finding that concentrations of aldehyde–nucleotide adducts correspond to free aldehydes both corroborates this

model and demonstrates a functional significance. EAC has a particularly high mutation rate, comprised of unique patterns representing incompletely explained etiologies[27]. Our study proposes aldehyde stress as a source of EAC genotoxicity. Gastro-esophageal acid reflux is a risk factor for EAC[29], and low pH enhances carbonyl-based nucleophilic addition reactions such as aldehyde–nucleotide adduction. Compared to the stomach, the esophageal pre-epithelial mucous layer is comparatively thin and contains no bicarbonate buffer[30], and so the lower esophagus may be particularly vulnerable to pH-facilitated aldehyde toxicity. Thus, the finding of aldehyde-DNA damage in EAC carries clinical impact, as *ALDH*-enhancing and aldehyde-scavenging agents exist which augment aldehyde detoxification (exemplified by metformin in this work)[31], with potential implications for chemopreventative strategies in the pre-EAC metaplastic condition, Barrett's esophagus.

We found decanal accumulation to be an affiliated metabolic phenotype of chromosome 17p deletion in EAC, which involves *TP53* deletion. Chromosome 17p loss-of-heterozygosity is associated with reduced survival[26], as were reduced *ALDH3A2* and increased decanal in this study. This may broaden the clinical application of exhaled aldehyde testing to EAC prognostication. Enhanced tumorigenecity from co-deleted *TP53* neighbors has brought 17p arm deletions into focus[32], and our results suggest that *ALDH3A2* could provide *TP53*-independent metabolic surrogates of this event. In breast cancer, lipid reprogramming following *CKND2*-driven chromosome 8p deletion has been identified[33]. Given that up to a quarter of a typical cancer genome involves arm-level copy number aberration[34], druggable genetic driver events may also provide reliable metabolic surrogates amenable to non-invasive testing, via collateral effects on non-redundant metabolic genes.

A recent pan-cancer analysis has found that expression of the *ALDH* isoenzymes differs greatly between cancers[35]. A number of context-specific factors may mediate the net phenotype from deregulation of an individual *ALDH* gene, including (i) the activity of the other *ALDH* genes, particularly those with overlapping specificities, (ii) the redox state, (iii) microenvironmental factors, including hypoxia and co-factor availability, (iv) mechanisms of endogenous aldehyde production, and (v) the activity of collateral detoxification and repair mechanisms, for example, the DNA damage response. This may explain why *ALDH2* was not significantly different in this study, despite underactive *ALDH2* being associated with ESCC, a different type of esophageal malignancy[10]. In addition, increased expression of *ALDH1A1*, *1B1*, *3A1* and others has been identified in other malignancies, with inverse associations with survival[14,36,37]. The oncogenic function has been attributed to stemness (*1A1*) or chemotherapy resistance (*3A1*), and whether these changes are causal or reactive is unclear. In this study, *ALDH1A3*, *-3A1*, *-4A1*, and *9A1* were under-expressed in both Barrett's epithelium and EAC, and the genotoxic aldehydes they detoxify were enriched. This suggests these phenotypes contribute to oncogenesis, and is in keeping with sequencing evidence of high mutational burden in both Barrett's epithelium and EAC[27,38]. In contrast, *ALDH3A2* expression loss was seen in EAC only, possibly as a consequence of tumor biology (i.e. *TP53* copy loss), rather than the cause.

A potential limitation is the finding that decanal is associated with more advanced disease stage, which is at odds with its inclusion in a breath test for early cancer detection. However, all studies demonstrating enriched decanal in EAC have been in UK cohorts derived from tertiary care centers, which are inherently enriched for treatable disease. EAC has dismal survival rates because most patients cannot be offered curative treatment, and the breath test (five biomarkers, including decanal) can detect treatable disease stages. In addition, we identified a positive stage correlation, which confirms this metabolite's relevance to disease biology. Changes in exhaled aldehydes and related compounds have been observed in other cancers[39], and how these biomarkers perform in different disease contexts is a research priority. The lack of difference in decanal concentration in diabetics taking metformin is unsurprising as decanal is relatively unreactive, which further supports its potential as a volatile biomarker.

Another limitation is that the present work aimed to understand events at a tissue level, rather than unpick the kinetics of these compounds to exhaled breath. Our previous work suggested hematological kinetics for fatty acid VOCs[40]. For aldehyde VOCs, the kinetics may be hematological or endoluminal. Related work has recently identified characteristic changes in the EAC phospholipid profile[41], which could be source molecules for endogenous aldehydes[42]. This will permit stable isotope tracing studies to demonstrate the kinetics of these compounds to the exhaled breath. It will also provide a platform to explore how potential influences such as source compound concentrations, mechanisms of production, and microenvironmental factors contribute to the overall aldehyde phenotype, in addition to detoxification and repair.

In summary, loss of aldehyde detoxification leads to endogenous aldehyde accumulation in EAC, generating genotoxins and more stable alkanals that could be a source for the same compounds detected in the exhaled metabolite pool. Our combined analytical and molecular approach is the first integrated investigation of tissue aldehyde metabolism in any cancer. Further molecular studies should seek to validate these findings, identify these compounds' parent metabolites, trace the kinetics to the breath, and establish whether endogenous aldehydes contribute to other upper gastrointestinal malignancies. In addition, these data also support the continued clinical investigation of exhaled aldehydes for EAC early diagnosis.

## Methods

**Patients**. Fresh human material and associated clinical metadata was collected and accessed under UK National Research Ethics Service Ref: 14/LO/0742, Imperial College Healthcare Tissuebank Review Board Approvals R14067, R14087, and R16018. Archived paraffin embedded tissue was accessed under 14/LO/0742, approval R14067, and through 10/H0305/1. Written informed consent was taken from all patients.

Patients were identified via the North West London Oesophago-gastric (OG) Cancer Unit (St. Marys' Hospital, Paddington, London, UK) and associated endoscopy department. Tissue samples were collected at endoscopy using 3 mm biopsy forceps thoroughly cleaned in UPLC-grade water. Typically 4–6 biopsies were harvested at 2 cm intervals per tissue type, separated circumferentially, to capture clonal diversity. EAC samples were collected from Siewert 1 or 2 EAC cases that had been confirmed histologically. Tumor-adjacent normal squamous (SqT) samples were collected 5 cm proximal to the proximal extent of the EAC or associated Barrett's, whichever was higher. Normal esophageal samples (SqN) were collected 5 cm proximal to the gastro-esophageal junction in control subjects without endoscopic abnormalities; prior or current diagnosis of cancer; active infection; or previous esophago-gastric surgery (e.g. gastric bypass). No restriction was placed on history of heartburn, proton pump inhibitor use, and alcohol and tobacco use. Biopsies were flash frozen in liquid nitrogen and stored at −80 °C. Matched biopsies were taken from all tissue sites for histopathologic confirmation. Biopsies were flank cryosectioned and checked histologically prior to homogenization (cellularity > 95%), if permissible by the methodology. All patients were starved for at least 6 h prior to sampling. Demographic details were compiled prospectively and listed in Supplementary Data 1.

**Cell lines and treatments**. Supplementary Data 1 lists the materials used in the study. Commercial EAC cell lines were acquired from European Collection of Authenticated Cell Cultures (via Public Health England) or the American Type Culture Collection at the start of the study and cultured according to instructions. Primary keratinocyte cultures were established using a rho-kinase method developed for skin keratinocytes[16]. Briefly, three endoscopic biopsies from patients with normal endoscopic appearances were pooled and minced in serum free Dulbecco's modified Eagle's media (DMEM; ThermoLife) containing gentamicin/amphotericin (Invitrogen). The homogenate was incubated in collagenase III (Sigma) for an hour at 37 °C, neutralized in DMEM containing 10% FBS and the antibiotics, and forced through a 100 μm mesh. The cells were

moved to fresh complete DMEM containing 10 μM Y-27632 (Sigma), and seeded onto plates coated with 0.01 μg/mL bovine fibronectin and bovine albumin and 0.03 μg/mL rat tail collagen (all Sigma), in the presence of irradiated 3T3-J2 feeder cells. Lines were continually passaged for 6 weeks and checked for mycoplasma twice prior to experiments. Y-27632 and feeder cells were withdrawn 2 weeks before experiments and fibroblast contamination was checked by immunoblotting for vimentin. Hypoxia experiments were conducted in a nitrogen incubator using otherwise standard culture conditions. All lines had a passage limit of 20 and were tested for mycoplasma monthly.

Diethylaminobenzaldehyde (Sigma) was dissolved in dimethylsulfoxide (DMSO) and was incubated with cells for 72 h. The DMSO concentration was kept constant (0.1%). RNA interference was achieved with oligofectamine (ThermoLife) according to the manufacturer's instructions and confirmed by immunoblotting. The total transfected RNA was kept constant across the conditions, supplementing with non-targeting RNA as needed. Five-day proliferation assays were in 96-well format using the 3-(4,5-dimethylthiazol-2-yl)-2,5-diphenyltetrazolium bromide (MTT) assay (ThermoLIfe). Reduced and oxidized glutathione levels were calculated using a fluorescence-based kit (Promega V6611). Cells were pre-treated with metformin (Sigma) for 48 h, including refreshed media after 24 h, and consistent metformin concentrations maintained for the aldehyde incubation. Concentrations for the formaldehyde and acetaldehyde challenge experiments were selected based on literature evaluation[43,44], and were within the measured concentrations from the tissue LC-MS data in the present study. To model the continuous production of endogenous aldehydes seen in situ, media containing fresh dilutions of the corresponding aldehydes was refreshed twice during incubation.

### Aldehyde quantification by ultra-performance liquid chromatography tandem mass spectrometry

*General principles.* Quantification of aldehydes from tissue samples used the dinitrophenylhydrazine method[45] with numerous updates. Fastidious measures were taken to minimize environmental contamination, including working in clean air environment, extracting DNPH (0.2 M in 70% ethanol in phosphoric acid, Sigma) four times with hexane (1:20 v/v) immediately prior to use, scrupulously clean materials which were baked at 70 °C overnight, and fresh bottles of UPLC-grade solvents were used for every run. Blank samples involving every element of the sample preparation process were regularly tested to ensure background was controlled and that carryover did not occur. Alternating low and high calibration points were injected regularly to ensure consistent instrument response.

Forty-two aldehydes and ketones were selected for study (as indicated in Supplementary Data 1). The rationale for analyte selection was any of: (i) known genotoxin, (ii) physiological metabolite, (iii) present in exhaled breath, (iv) a specific target for ALDH isoenzymes of interest. Analytical variation was controlled with six isotope-labeled standards (ISTDs)—C2:0-d₄ (Sigma, for short-chain alkanals), C6:0-d₁₂ (Sigma, for medium-chain alkanals), C16:0-d₅ (Santa Cruz, for fatty alkanals), HNE-d₃ (Cambridge Bioscience, for dienals), ONE-d₃ (Cambridge Bioscience, for dialdehydes), and MDA-d₂ (prepared by diluting 10 μL 1,1,3,3-tetraethoxypropane-1,3-d₂ (Santa Cruz) in 10 mL of 0.1 M HCl and hydrolyzing at 100 °C for 10 min)[46].

*Sample preparation.* For tissue, three endoscopic biopsies (9–12 mg) were homogenized under liquid nitrogen in a clean pestle and mortar and weighed. To this 180 μL of dry-ice-cold UPLC acetonitrile/water (50:50 v/v) containing 62.5 ng/mL internal standards was added with a baked ceramic bead. The slurry was extracted in a Reitsch oscillator for 60 s at −40 °C, and protein cleared by centrifugation (20,000*g*, 3 min). The supernatant was derivatized with 40 μL DNPH; the pellet re-extracted with the same volume of extraction solvent. The two extracts were combined and derivatized for 1 h at 25 °C. The phases were separated by adding 20 mg of baked sodium chloride, followed by centrifugation at 20,000*g*. The organic phase was transferred to a clean glass UPLC vial and sealed with Teflon. In vitro aldehydes were assayed by mixing 180 μL of media with 180 μL acetonitrile containing 62.5 ng/μL ISTDs, which was then centrifuged, and derivatized as above. Protein content was measured per well by the bicinchoninic acid method (Sigma). Sample preparation for each experiment was conducted in a continuous run on a single day.

*UPLC-MS/MS analysis.* This was undertaken on a Acquity UPLC and Waters TQS MS/MS system. A C18 Cortecs column (particle size 1.6 μm, internal pore 2.1 μm, Waters) was used, with UPLC-grade water (A) and UPLC-grade acetonitrile (B) (both Sigma) mobile phases. Column temperature was 40 °C, sample temperature 4 °C, and the flow rate 0.5 mL/min. Injection volume was 5 μL, under initial conditions of 30% B. The ratio was changed as follows: 0.20–8 min to 40%, then 8–16 min to 95%, 19–19.1 min to 30% and held until 20 min. The source settings were as follows: source temperature 150 °C, capillary voltage 2.5 kV, cone voltage 10 V, cone gas flow rate 200 L/h, desolvation gas temperature 400 °C, desolvation gas flow rate 650 L/h. A scheduled multi-reaction monitoring MS/MS method was established (Supplementary Data 2). Each sample was prepared in 2–3 biological replicates depending on availability, with the final result being averaged (one result per tissue for each patient in the final analysis).

*Data processing.* Acquired peaks were integrated using Targetlynx (Waters, SCN855) and manually checked. For quantitative analysis, internal standard calibration curves were calculated by dividing a serially diluted mix of all unlabeled standards by the respective ISTD (concentration 50 ng/μL, see Supplementary Data 1). The intensities of the unknown concentrations were also divided by the respective ISTDs, and concentrations calculated by comparing to the ISTD calibration. Tissue aldehydes were normalized to sample mass (milligrams). In vitro aldehydes were normalized to protein concentration in a parallel well as quantified by the bicinchoninic acid method. Isomer/isobar ketones, alkanals, and dialdehydes (e.g. acetone, propanal, glyoxal, molecular weight = 58) were unambiguously determined on the basis of retention time, parent ion mass-to-charge (e.g. dialdehydes reacting to form a di-DNPH hydrazine) and characteristic abundant fragment ions (e.g. 163, 152, 182 for alkanals, ketones, and dialdehydes, respectively). A lower limit of quantification (LLOQ) of 20 pg/mg was set, given a minimum tissue input of 10 mg, an extraction volume of 0.2 mL, and that 1 ng/mL was well above the limit of detection for all targets. The manually checked integrations were exported to R to reformat the matrix (RStudio version 1.1.456, RStudio, Inc., code provided in Supplementary Data 7) and complete the processing as above. All data processing was blinded.

### Aldehyde quantification using proton transfer reaction time-of-flight mass spectrometry

Measurements were conducted employing a commercial PTR-MS instrument (PTR-TOF 1000; Ionicon Analytik GmbH, Innsbruck, Austria). Drift tube conditions were temperature 110 °C, pressure 2.30 mbar, and voltage 350 V, resulting in an E/N of 84 Td (1 Townsend = 10–17 V cm²). Optimal measurement conditions were chosen based on a validated experimental workflow[47]. Sample inlet flow was set to 40 sccm. During the described experiments, a series of daily quality checks were conducted on the PTR-TOF-MS: we assessed the amounts of spurious ions with the two ionization modes used ($H_3O^+$ and $NO^+$), and we measured accuracy on a benzene certified standard and fragmentation on butanal and butyric acid. Mass resolution was checked daily and optimized whenever needed. Tissues samples (10–20 mg) were thawed in glass 20 ml headspace vials with Teflon insert screw-caps (ThermoLife) left to equilibrate at 25 °C for 30 min. The headspace was sampled for 10 s at 40 mL/min with isobaric volume replacement with cylinder air. All data analysis was carried out using PTR-MS Viewer 3.2.2.2 (Iconicon), following the manufacturer's instructions. Accurate mass peaks reaching a signal-to-noise ratio >10:1 were integrated and normalized to the total ion emission. This procedure accounted for sample-to-sample variation in mass and surface area.

### Aldehyde–nucleotide adduct quantification by ultra-performance liquid chromatography tandem mass spectrometry

Two aldehyde adducts of purine nucleotide were selected for quantification: 1-$N^6$-etheno-2′-deoxyadenosine (edA), and α-methyl-γ-hydroxy-1,$N^2$-propano-2′-deoxyguanosine (CrodG). For tissue samples, five endoscopic biopsies from SqN, SqT, or EAC collected and pooled. Leukocytes were extracted from the buffy coat of 10 mL blood collected from EAC patients. DNA clean-up was with the Gentra Puregene kit (Qiagen). Three isotope-labeled internal standards were used at a concentration of 5 ng/mL: deoxyadenosine-$N^{13}_5$ (Cambridge Isotopes), 1-$N^6$-etheno-2′-deoxyadenosine-$N^{13}_5$ (produced by reacting chloroacetaldehyde with deoxyadenosine-$N^{13}_5$)[48], and α-methyl-γ-hydroxy-1,$N^2$-propano-2′-deoxyguanosine-$^{13}$C,$^{15}$N$_2$ (Toronto Research Chemicals). UPLC-MS/MS and data processing were as previously described[19].

**Bioinformatics.** Two datasets, GSE26886 and GSE13898 were found on ArrayExpress and Gene Expression Omnibus reporting esophageal squamous mucosa and EAC transcriptomes and were included in geneset enrichment analysis (Broad Institute, MIT, Cambridge, MA, USA)[49]. The default settings and the Gene Ontology geneset set (version 4.0) were selected for the analysis. GSE26886 used laser capture microdissection for tissue isolation and was thus selected for presentation in Fig. 2. To identify candidate genes in the aldehyde oxidoreductase geneset, a univariate analysis of the datasets was conducting by comparing groups with a Student's *t*-test, using a Bonferroni significance threshold of $P = 10^{-6}$. GSE26886 was also selected for Ingenuity Pathway Analysis (Core Analysis, Qiagen) using standard settings.

Legacy data from The Cancer Genome Atlas (available at https://portal.gdc.cancer.gov/projects/TCGA-ESCA) and the International Cancer Genome Consortium ICGC (available at https://ega-archive.org/studies/EGAS00001000725) were analyzed for *ALDH3A2* regulatory hypotheses (last date accessed: 1 June 2019). TCGA-ESCA data are a mixed of EAC and squamous cell cancers, and so EAC cases (*n* = 87) were selected in these analyses, and a subset of these had survival data (*n* = 57). ICGC data from OCCAMS were limited to 85 cases that had a predicted normal-cell contamination <50%. For each case germline SNPs and indels were called using the GATK (3.2-2) HaplotypeCaller, and reads supporting each allele at germline heterozygous positions were counted for both the tumor and matched normal. Copy number alterations along with ploidy and purity estimates were then derived from these read counts using ASCAT (2.3), and compared to matched RNA-seq expression data.

**mRNA quantification.** MIQE guidelines were adhered to for all polymerase chain reaction (PCR) experiments[50]. For in vivo *ALDH* expression analyses, a sample size

was determined from GSE26886 expression distributions, taking $\alpha = 0.05$ and $\beta = 0.8$, which returned 5 (ALDH4A1) to 67 (ALDH1A1) paired samples. Endoscopic tissue biopsies were flank cryosectioned and microdissected if necessary to achieve a cell purity >90%. Homogenization was in Trizol (ThermoLIfe) using three-step homogenization (manual pestle grinding for 2 min, ceramic bead-beating for 3 min, and then Qiashredder (Qiagen)). RNA was fractionated and then purified (RNAeasy, Qiagen). Reverse transcription was with Superscript III kit (Thermo-LIfe), and quantitative PCR using PowerSybr PCR master mix (ThermoLife). Primers were designed in Primer3 (see Supplementary Data 6). The MIQE-suggested panel of 10 reference genes was assayed in a panel of 10 random samples and HPRT1 was found to be the most stable. All data processing was blinded.

**Immunohistochemistry.** We collated tissue microarrays or whole-mount sections from 571 EAC patients from Imperial College Healthcare NHS Trust tissuebank and national collaborative resources (the OCCAMS and POEM collaborations, see "Acknowledgements"). The Leica Bond™ system was used. Sections (7 µm) were deparaffinized, hydrated, and then heat mediated antigen retrieval was performed in citrate-based pH 6.0 solution for ALDH3A1 staining and EDTA based pH9.0 solution for ALDH3A2. The endogenous peroxidase was quenched with 3% hydrogen peroxide. The sections were incubated ALDH3A1 antibody (1:200 dilution) or ALDH3A2 antibody (1:200 dilution) and subsequently incubated with anti-rabbit IgG conjugated with polymeric horseradish peroxidase linker (Leica Bond Polymer Refine Detection, DS9800). DAB was used as the chromogen and the sections were then counterstained with hematoxylin and mounted with DPX. Antibodies were as for immunoblotting and details are provided in the Supplementary Data 1.

Sections were imaged (NanoZoomer 2.0HT, Hamamatsu). Scoring was by two pathologists who were blinded to the metadata. For whole-mount sections SqT, BAR, and EAC regions were scored in five random high-powered areas according 0–3 on a basis of staining intensity and prevalence (0 = no staining or <50% mild staining; $1 \geq 50\%$ mild staining, no moderate staining; 2 = any moderate staining, <50% strong staining; $3 \geq 50\%$ strong staining)[51] with a single average score per tissue type per patient used for comparative analysis. For correlation to metadata, the immunoscore was dichotomized, with a cut-off of >1 being positive. For tissue microarrays, replicate cores for each patient were provided in quintuplet (Imperial array) or triplicate (OCCAMS and POEM arrays, see "Acknowledgements"). Averaging across scorers and replicates generated a single score per patient.

**Immunoblotting.** Protein samples (30 µg) were separated by SDS-PAGE and transferred to PVDF membranes (all Biorad). Blots were probed with antibodies listed in Supplementary Data 1. The following antibody dilutions were used for immunoblotting: ALDH1A3 1:333, ALDH3A1 1:500, ALDH3A2 1:333, ALDH4A1 1:2000, ALDH9A1 1:500, Tubulin 1:2000, Cyclophilin B 1:1000, E-cadherin 1:1000, CK5/6 1:2000, p63 1:2000, Phospho-Histone H2AX (Ser139) 1:1000, TP53 1:2000, Vimentin 1:2000, Phospho-ATM (Ser1981) 1:1000, Phospho-ATR (Ser428) 1:1000, Phospho-CHK1 (Ser345) 1:1000, and Phospho-CHK2 (Thr68) 1:1000.

**Fluorescence in situ hybridization.** The following BACs were used as probes for the FISH experiments: RP11-227J24, containing alpha satellite specific for the centromere of chromosome 17; RP11-352K5 (hg19 17:19,532,515–19,692,855, ALDH3A1 and -3A2 (SourceBioscience); RP11-89D11 (hg19 17:7,495,711-7,663,042, TP53 (SourceBioscience). The BACs were labeled according to the manufacturer' instructions, using the Nick Translation System kit (Abbott Molecular), incorporating ChromaTide Alexa Fluor 488-5-dUTP (Thermo Fisher Scientific), ChromaTide Alexa Fluor 594-5-dUTP (Thermo Fisher Scientific), or biotin-16-dUTP (Sigma), respectively. The probes were re-suspended in hybridization buffer (50% formamide, 10% dextran sulfate, 2× SSC) at 10 ng/µL, in the presence of a 10× excess of unlabeled human Cot DNA (Sigma).

The triplicate TMAs from the Bristol and Oxford OCCAMS cohorts were used as their local fixing process generated minimal background fluorescence. The slides were dewaxed and hydrated in xylene and ethanol respectively. Heat-induced epitope retrieval was in TE buffer, at 95 °C, for 15 min, then briefly equilibrated in distilled water at room temperature. Protein digestion was carried out at 37 °C, using Digest All-3 pepsin (Thermo Fisher Scientific). The cells were briefly dehydrated in ethanol 96%, and then 20 µL of probe mix were applied to the slides, under a glass coverslip. Following denaturation at 85 °C for 5 min, the cells were incubated overnight at 37 °C. Post-hybridization washes were carried out at 60 °C in 0.1× SSC. The biotinylated probe was detected using streptavidin-647 (Thermo Fisher Scientific). The Papworth and Glasgow arrays were not included in the analysis as the DNA binding sites could not be retrieved, presumably through differences in the original histological preparation.

The slides were mounted in DAPI/Vectashield (Oncor), and analyzed using the Delta Vision Elite Imaging System. For each core, 2–3 high cellularity (>95% adenocarcinoma) fields were acquired as 35 × 0.2 µm z-stacks, and the resulting images were deconvolved. The stacks were compressed using the z-projection function in Fiji (Fiji Is Just Imagej)[52], generating average projection images (DAPI) and max projection images (for the three colors). A fully automated image analysis method was constructed in Cell Profiler (Broad Institute). Masks were generated using the DAPI image, selecting nuclei on size and roundness to filter out non-

malignant cells, improving cellularity to >99% (average of 20 images, manual counting). The average counts per mask were calculated for each probe, and typically 300–500 masks were counted per field. Target probes count (ALDH3A1/2 and TP53) were presented as a function of the control probe count (centromere 17).

**Quantification and statistical analysis.** Calculations for bench experiments were conducted in Metaboanalyst 4.0 (https://www.metaboanalyst.ca/), Prism (version 7.0), SPSS (version 26) or R. For principal component analysis, measured aldehyde concentrations were adjusted by weight, and then mean-centered and log-transformed to normalize the data and give each feature equal weight in the model. Prism or GENE-E (Broad Institute) was used to visualize metabolic data as heatmaps. All experimental groups showed similar variances. Mann–Whitney U-test or one-way ANOVA with Bonferroni correction was used to compare groups, *$P < 0.05$, **$P < 0.01$, ***$P < 0.001$, ****$P < 0.0001$. Further details are provided in the figure legends.

For the multivariable diagnostic model based on tissue aldehyde concentrations, we applied our previous regression-based methodology to select features[6]. Briefly, a binary logistic regression model was fitted, using squamous samples from healthy patients (SqN) as the control group, and any sample from the malignant esophagus as the test group (SqT, EAC, BAR). A stepwise approach was used to add significant features until the model could no longer be improved. These analyses were conducted in SPSS. Metabolites contributing to the final model were then used to create a multivariable ROC curve using the Random Forests approach, using the Biomarker package of Metaboanalyst.

**Reporting summary.** Further information on research design is available in the Nature Research Reporting Summary linked to this article.

## Data availability

Legacy microarray expression studies re-analyzed here (in Fig. 1 and Supplementary Fig. 1) are available at https://www.ncbi.nlm.nih.gov/gds, under the accession codes: GSE26886, GSE13898, GSE39491) and GSE34619. Data from The Cancer Genome Atlas (used in Fig. 5 and Supplementary Fig. 6) is available at https://portal.gdc.cancer.gov/projects/TCGA-ESCA. Data from the International Cancer Genome Consortium (used in Supplementary Fig. 6) is available at https://ega-archive.org/studies/EGAS00001000725. Data from The Cancer Cell Line Encyclopedia (used in Supplementary Fig. 6) is available at https://portals.broadinstitute.org/ccle. Source data are provided with this paper.

## Code availability

The code pertaining to the transfer of Targetlynx peak integrations to R is provided in Supplementary Data 7.

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

## Acknowledgements

This research was supported by the National Institute for Health Research (NIHR) London In-Vitro Diagnostic, NIHR-Biomedical Research Centre, the Imperial College Charity, and the Rosetrees and Stoneygate Trusts. The Chromosome Dynamics Core is supported by the Wellcome Trust (award 203141). The views expressed are those of the authors and not necessarily those of the Wellcome Trust, the NHS, the NIHR or the Department of Health. The laboratory of R.C.F. is funded by a Core Programme Grant from the Medical Research Council (RG84369). Whole-genome sequencing was funded by a program grant from Cancer Research UK (RG81771/84119). The authors thank collaboration members for access to genomic data, tissue microarrays, and associated metadata. The following OCCAMS members and centers contributed: P.A.W. Edwards, N. Grehan, B. Nutzinger, C. Hughes, E. Fidziukiewicz, S. MacRae, A. Northrop, G. Contino, X. Li, R. de la Rue, A. Katz-Summercorn, S. Abbas, D. Loureda, M. O'Donovan, A. Miremadi, S. Malhotra, M. Tripathi, S. Tavaré, A. G. Lynch, M. Eldridge, M. Secrier, G. Devonshire, S. Jammula (University of Cambridge); J. Davies, C. Crichton (University of Oxford); N. Carroll, P. Safranek, A. Hindmarsh, V. Sujendran (Cambridge University Hospitals NHS Foundation Trust); S. J. Hayes, Y. Ang (Salford Royal NHS Foundation Trust); A. Sharrocks (University of Manchester); S. R. Preston, S. Oakes, I. Bagwan (Royal Surrey County Hospital NHS Foundation Trust); V. Save, R.J.E. Skipworth, R. O'Neill (Edinburgh Royal Infirmary); T. R. Hupp (Edinburgh University); O. Tucker, A. Beggs, P. Taniere, S. Puig (University Hospitals Birmingham NHS Foundation Trust); T. J. Underwood, R. C. Walker, B.L. Grace (University Hospital Southampton NHS Foundation Trust); H. Barr, N. Shepherd, O. Old (Gloucester Royal Hospital); J. Lagergren, J. Gossage, A. Davies, F. Chang, J. Zylstra, U. Mahadeva (Guy's and St Thomas's NHS Foundation Trust); V. Goh, F.D. Ciccarelli (King's College London); G. Sanders, R. Berrisford, C. Harden (Plymouth Hospitals NHS Trust); M. Lewis, E. Cheong, B. Kumar (Norfolk and Norwich University Hospital NHS Foundation Trust); S.L. Parsons, I. Soomro, P. Kaye, J. Saunders (Nottingham University Hospitals NHS Trust); L. Lovat, H. Haidry (University College London); M. Scott (Wythenshawe Hospital); S. Sothi, S. Suortamo (University Hospitals Coventry and Warwickshire NHS); S. Lishman (Peterborough Hospitals NHS Trust); C.J. Peters, K. Moorthy (Imperial College London); A. Grabowska (University of Nottingham); R. Turkington, D. McManus, H. Coleman (Queen's University Belfast); D. Khoo, W. Fickling (Queen's Hospital, Romford). The following POEM members and centers contributed: R. O'Neill, R. Skipworth (Royal Infirmary of Edinburgh); J. O'Sullivan, N. Lynam-Lennon, A. Cannon, J. Reynolds (St. James Hospital, Dublin). The

authors also thank C. Rajaguru, N. Imrit, N. Maynard (Oxford University NHS Foundation Trust, Oxford); J. Going, M. McKernan R. Stuart (NHS Greater Glasgow and Clyde, Glasgow); M. Moorghen, J. Blazeby, C.P. Barham (Bristol Royal Infirmary, Bristol); P. Vlavianos, J. Hoare, I. Belluomo, A. Myridakis (Imperial College London).

## Author contributions

S.A. designed research, carried out experiments, analyzed data, and wrote the paper. Z.B. carried out the LC-MS experiments. T.W., S.R.M., M.E.A., Y.M.G., C.J.P., and P.R.B. provided samples, collated metadata, analyzed data, and carried out the PTR-MS experiments. H.L., N.H., and H.G. oversaw the molecular experiments. H.K. carried out the immunohistochemistry experiments. F.R. and R.G. analyzed immunohistochemistry experiments. D.M. and C.G. carried out the FISH experiments. R.C.F. oversaw re-analyses of genomic data and study design. Z.T. oversaw the LC-MS experiments and the research design. G.B.H. conceived the study, directed the research, and co-wrote the paper. All authors agreed to the final version of the paper.

## Competing interests

G.H. is named on patents related to breath analysis, and is a shareholder in VODCA, a cancer detection company. R.C.F. is named on patents related to the Cytosponge and associated assays and is a shareholder and consultant for Cyted an early detection company. The other co-authors declare no competing interests.
