## [Peer Review File · Nature Communications]

Reviewers' comments:

Reviewer #1 (Remarks to the Author):

In this study, the authors present data that volatile aldehydes are enriched in esophageal adenocarcinoma (EAC) patients' breath, due to decreased aldehyde dehydrogenase expression in tumor tissue, and this could improve early diagnosis. In general, this is a well-designed study with very interesting data that may have a translation impact. However, there are several points that the authors need to address:

- One basic question is how VOC from the esophagus end up in the exhaled breath? What is the mechanism and what are the appropriate controls for this?
- A second basic argument is that the authors' data, although very interesting, may not have a prognostic value.
- A third but important question is whether the findings regarding the link between decreased ALDHs and increased aldehydes are the results of the tumor biology/condition or the cause?
- Since ALDH3A2 is down, there must be a better explanation why there is not difference in the fatty aldehydes as seen for the short-chain alkanals (forming DNA-adducts) and medium chain (alkenals, decanal).
- Decanal has been identified in the exhaled breath of patients with colon cancer PMID: 28777343.
- The doses used for formaldehyde and acetaldehyde are not realistic nor physiologic. The effect on H2AX are seen in the 1 (minimal) and 5 mM range (Figure 3) that you will not see it in any normal or pathologic condition.
- Formaldehyde is metabolized by an alcohol dehydrogenase.
- The authors argument that their data are consistent with epidemiological studies that linked inactivating ALDH variants to esophageal malignancies, is rather weak and not aligned with their data. The references provided are for the ALDH2 and ALDH1A2 genes. No differences in ALDH2 expression are noted in their data.
- Based on the published data, there is no indication that patients with Sjogren-Larsson syndrome, caused by homozygous or compound heterozygous mutation in the ALDH3A2 gene which encodes fatty aldehyde dehydrogenase, have any incidence of esophageal cancers.
- Metformin, a listed as direct aldehyde scavenger but the paper cited (#11) mention a possible interaction between aldehydes and metformin.
- In Figure 2, the formaldehyde, 2-butanol and acetaldehyde data are not in agreement 4-HNE (genotoxic) data with the other data from medium chain aldehydes.
- In most of the cases, high expression of ALDHs, especially ALDH1A1, ALDH3A1 and ALDH1B1 is associated with various cancers and genetic abolishment of these genes is associated with decreased cancer formation (e.g., PMID: 28881356, 26566217; 29767973, 28280079, 29767973). The authors should comment this as their results are the opposite

Overall, it is a highly sophisticated paper in terms of methodology, approaches and clinical/translational significance. The authors should improve the biology of the paper especially in the discussion.

Vasilis Vasiliou

Reviewer #2 (Remarks to the Author):

I have a few comments, which I think the authors should address.

Comment 1: to me the most important related to details of the pathological tumour staging of the EAC series from which the obtained breath sampled for breath analysis. Hence In other words 8th edition AJCC/UICC staging of EACs (esophagogastric junction cancers), the authors should indicate to the non-clinical scientific reads the spread in terms of the pathological staging of the tumours in their data base outlined simply below:

Tis High-grade dysplasia, defined as malignant cells confined by the basement membrane
T1 Tumor invades the lamina propria, muscularis mucosae, or submucosa

T1a* Tumor invades the lamina propria or muscularis mucosae

T1b* Tumor invades the submucosa

T2 Tumor invades the muscularis propria

T3 Tumor invades adventitia

T4 Tumor invades adjacent structures

The value of the study would depend on the percentage of T2, T3 and T4 as opposed to Tis, T1, T1a, T1b

Comment 2: The study data showed that decanal is associated with more advanced tumour stage of EAC. Is this mirrored in the previous breath analysis study?

Comment 3: Availability of underpinning technology (ultra-performance liquid chromatography tandem mass spectrometry) for measuring expired aldehydes. I am certain that the technology is unlikely to be available outside the large University Medical Schools, e.g., GP practices and district general hospital. Hence would there be an NHS reference laboratory for breath analysis of aldehydes and other volatile compounds?

Reviewer #3 (Remarks to the Author):

The manuscript titled "Endogenous aldehyde accumulation generates genotoxicity and exhaled biomarkers in esophageal adenocarcinoma" focuses on describing and characterizing aldehyde biochemistry in the esophagus and how this may serve as a non-invasive tool for detecting esophageal adenocarcinoma (EAC). The manuscript is generally well written and the research topic is of strong interest to the scientific community. The research builds logically and incrementally on earlier work by the research team linking breath aldehyde to tissue and conducting limited in vitro work utilizing human EAC cell lines. However, portions of the data and results lack clarity, are not convincing and suffer from reduced scientific rigor. Thus, despite some interesting positive preliminary data, some concerns require resolution for the research findings to be fully appreciated.

Specific areas for improvement:

1. Abstract, it is premature to claim the results support aldehyde metabolism as a chemoprevention target given the results to date have been largely in late stage EACs and discerning between Barrett's progressors and non-progressors for example or BAR and EAC remains unclear. This comment applies to the discussion as well.
2. Intro, line 74, ref 4 is from 2008, may consider updating the reference as it seems more current data supports that the increase in EAC incidence has leveled off in recent years. Also, pertains to 348. Alternatively add context.
3. The justification for the choice of aldehyde related markers chosen for evaluation is summarized in Supp Figure 1. Specifically, Fig 1b is the basis for discerning differences across publicly available data sets linked to pathology. The authors identified ALDH1A3,-3A1,-3A2,-4A1 and -9A1 as under-expressed in EAC or glandular tissue compared to esophageal squamous mucosa. "Normal" should be added to ... eso sq epithelium, line 107 and it should be made clear that based on supp Fig 1b both Barrett's and EAC under-express ALDH genes compared to normal eso.
4. Supp Figures 1d and 1e appear to have the very same/duplicated loading control bands which is not scientifically possible. Tubulin runs between 50 and 55 kDa making it nearly impossible to be the real loading control for some of the proteins assessed which run at similar kDa. Representative images of loading controls are understood, but duplication is not acceptable.
5. Scale/Enlarge figure 1C so it is legible upon printing.
6. Supp 1d and 1e compares EAC cell lines to keratinocyte cell lines, but not normal esophageal cell lines or Barrett's cell lines which would be informative and more relevant. Moreover, Barrett's cell lines are readily commercially available.
7. Supp 1C, study groups appear normalized to a loading control-GAPDH, thus, what SqT/EAC notation indicate relative to the plot shown in supp Fig 1C is unclear? Indicate the statistical test utilized for significance in Supp Fig 1C, as there is great variation in data; yet, highly significant p-values.

8. Description of ALDH3A results is confusing, line 119. The data that this discussion links to is unclear, please clarify.
9. Supp Fig 2 appears mislabeled as no Supp 2d is shown; yet, discussed in the text. Additionally, results referring to stabilization of HNE-protein adducts is not convincing. Sole use of OE33 cells may not reflect generalizable effects.
10. Figure 2 results are unclear in that some of the bars are not marked as indicating significance or NS making it difficult to interpret the claims in the text.
11. Supp Fig 4, Proper controls appear missing for Fig 4C. Why are the levels of metformin to much higher in sup fig 4d compared to sup 4c—is this a relevant concentration and why is yet a different concentration of metformin used in supp 4e. The inconsistency makes interpretation difficult as dose surely has an influence.
12. Most of the cell culture work is done in a single cell line and not consistent across experiments (i.e. viability and drug challenge in Flo1, siRNAtargets in OE33) making it difficult to understand whether effects noted are limited to the selected specific cell line versus a panel of BAR or EAC cells.
13. Fig 3 e, Western blot of poor quality/resolution and is not convincing. The blot should be repeated.
14. Despite the emphasis on decanal and claims made that decanal accumulation is affiliated with a metabolic phenotype there is no discussion of the rather opposing results showing that decanal does not differ in patients treated with metformin.
15. The claim that decanal can identify patients with treatable disease is overreaching.
16. Correct, Figure 5 is out of order, two 5g figures are shown.

REPLY TO REVIEWERS' COMMENTS

RE: Endogenous aldehyde accumulation generates genotoxicity and exhaled biomarkers in esophageal adenocarcinoma

Reviewer #1 (Remarks to the Author):

In this study, the authors present data that volatile aldehydes are enriched in esophageal adenocarcinoma (EAC) patients' breath, due to decreased aldehyde dehydrogenase expression in tumor tissue, and this could improve early diagnosis. In general, this is a well-designed study with very interesting data that may have a translation impact. However, there are several points that the authors need to address:

REVIEWER: One basic question is how VOC from the esophagus end up in the exhaled breath?

What is the mechanism and what are the appropriate controls for this?

AUTHOR RESPONSE: These metabolites' kinetics from tissue to the breath could either be haematological, or luminal via the unbroken pre-epithelial mucous layer of the oesophagus. We have previously published haematological kinetics for fatty acid VOCs from upper gastrointestinal cancer (Adam et al, Anal Chem 2019, PMID: 30699297).

However, for aldehyde VOCs, we favour luminal kinetics, based on the following evidence: (i) aldehyde metabolic field effects were identified in the UPLC-MS/MS study, with proximal normal epithelium adjacent to EAC also enriched for exhaled metabolites (controlled with normal epithelium from healthy volunteers, refer to Figure 2) (ii) the same aldehydes were enriched in the headspace of non-homogenised EAC tissue i.e. the tissue surface has an aldehyde vapor pressure (refer to Figure 4) (iii) our study of plasma aldehydes using UPLC-MS/MS did not identify significant enrichment in EAC samples (data available on request) (iv) blood from the gastro-oesophageal junction directly drains to the liver, which strongly expresses *ALDH* isoenzymes and should convert aldehydes to fatty acids.

Despite these suggestive points, proving the kinetic mechanism to the breath was beyond the scope of this work, which aimed to understand what was happening at a tissue level.

Since this study completed we have published candidate source compounds for the aldehydes (Abbassi-Ghadi et al, Cancer Research, 2020 PMID: 32345674). This will permit stable isotope labelling of the source molecules (with relevant controls) to establish the kinetics to the breath. We have therefore revised the discussion with the following lines (Page 18, lines 417-426):

“Another limitation is that the present work aimed to understand events at a tissue level, rather than unpick the kinetics of these compounds to exhaled breath. Our previous work suggested haematological kinetics for fatty acid VOCs⁴⁰. For aldehyde VOCs, the kinetics may be haematological or endoluminal. Related work has recently identified characteristic changes in the EAC phospholipid profile⁴¹, which could be source molecules for endogenous aldehydes⁴². This will permit stable isotope tracing studies to demonstrate the kinetics of these compounds to the exhaled breath. It will also provide a platform to explore the relative contribution of potential influences to the overall aldehyde phenotype, including source compound concentrations, mechanisms of production, and microenvironmental factors, as well as detoxification.”

REVIEWER: A second basic argument is that the authors’ data, although very interesting, may not have a prognostic value.

AUTHOR RESPONSE: We identified and validated low *ALDH3A2* and increased decanal as being associated with aggressive disease and poorer survival, in several datasets, using with complementary techniques (TCGA, immunohistochemistry, exhaled decanal, see Figure 5). These data suggest prognostic value, and stress how intrinsic this biology is to the disease phenotype. Please refer to (Page 16, Line 373):

“This may broaden the clinical application of exhaled aldehyde testing to EAC prognostication.”

REVIEWER: A third but important question is whether the findings regarding the link between decreased ALDHs and increased aldehydes are the results of the tumor biology/condition or the cause?

AUTHOR RESPONSE: We have revised the Discussion accordingly (Page 17, lines 395-401):

“In this study, *ALDH1A3*, *-3A1*, *-4A1*, and *9A1* were under-expressed in both BAR and EAC, and the genotoxic aldehydes they detoxify were enriched. This suggests these phenotypes contribute to oncogenesis, and is in keeping with sequencing evidence of high mutational burden in both Barrett’s epithelium and EAC^{27,39}. In contrast *ALDH3A2* expression-loss was seen in EAC only, possibly as a consequence of tumor biology (i.e. *TP53* copy loss), rather than the cause.”

By way of explanation, we believe the cause of *ALDH* deregulation is different for the individual isoenzymes, as is their contribution to oncogenesis. In tissue samples, *ALDH1A3*, *-3A1*, *-4A1*, and *9A1* were depleted in both Barretts epithelium and EAC. These isoenzymes have specificity towards genotoxic aldehydes. The net effect is altered redox, enriched genotoxic aldehydes, and direct DNA damage, however the precise metabolic phenotype is difficult to predict because of overlapping specificities and microenvironmental mediators. This hypothesis is supported by sequencing data, which demonstrates substantial mutational burden in both Barrett’s epithelium and esophageal adenocarcinoma, with the difference being a handful of changes in key driver genes (Weaver et al, Nature Genetics, 2014, PMID: 24952744; Frankell et al, Nature Genetics 2019, PMID: 30718927). Thus, for these isoenzymes, the expression changes are the inherent phenotype of Barrett’s cells, and this low expression may contribute to the development of invasive disease.

In contrast, *ALDH3A2* was only depleted in EAC in tissue samples, suggesting a link to progression (see Figure 1 and Supplementary Figure 1). We linked decreased *ALDH3A2* expression to loss of copy involving *TP53*, the only recurrent (>80%) driver gene for EAC. Thus, for *ALDH3A2* only, the expression change may be result of tumour biology. However, we have not found any evidence that this change is independently oncogenic.

REVIEWER: Since ALDH3A2 is down, there must be a better explanation why there is not difference in the fatty aldehydes as seen for the short-chain alkanals (forming DNA-adducts) and medium chain (alkenals, decanal).

AUTHOR RESPONSE: The concentration of fatty aldehydes was surprisingly very high, and the data was widely spread. When comparing proximal normal squamous and adenocarcinoma tissue from EAC patients, there was a trend toward enriched C16 and C18 in adenocarcinoma, however this did not pass multiplicity correction (see Supplementary Figure 3p-r). Similarly, when stratified by expression of *ALDH3A2* (presented for medium alkanals in Figure 5a,b), there was a trend toward enriched C16/18 in low expressing tumours, however this did not pass multiplicity correction. We feel that the surprisingly high concentrations of fatty aldehydes, and how they relate to *ALDH3A2*, warrants further focused study. However, ultimately we chose not to focus on the fatty aldehydes in this study, as they are inherently non-volatile and would not contribute to the parallel clinical translation.

REVIEWER: Decanal has been identified in the exhaled breath of patients with colon cancer PMID: 28777343.

AUTHOR RESPONSE: The cited review by Bhattacharya et al cites four papers assessing colorectal cancer exhaled VOCs. Of these, one (Altomare) found enriched decanal, whereas the other three did not find decanal despite appropriate analytics. We had methodological concerns about the Altomare approach including the controls used, and provided comment at the time of publication, which was accepted by those authors (PMID: 26583677). We also assessed colorectal cancer VOCs in larger cohorts, and did not find decanal to be a colorectal biomarker (Markar et al, Annals of Surgery, 2016 PMID:29194085). We have since confirmed that decanal is not a colorectal cancer breath biomarker, in the COBRA study (n=2500, NCT03699163, data submitted for publication). In contrast, exhaled decanal as an EAC biomarker has been validated in two large cohorts (Kumar et al, Annals of Surgery, 2013 PMID: 25575255 and Markar et al, JAMA Onc, 2019 PMID: 29799976). Despite that decanal has not been replicated in colorectal, we acknowledge that aldehydes including decanal do appear in isolated VOC studies of other cancers, and why this is the case requires further study. We have reviewed VOC aldehydes in other cancers (PMID: 30128487, see Supplementary) and provided this reference with a new line in the Discussion (Page 19, Lines 411-413):

“Changes in exhaled aldehydes and related compounds have been observed in other cancers³⁵, and how these biomarkers perform in different disease contexts is a research priority.”

REVIEWER: The doses used for formaldehyde and acetaldehyde are not realistic nor physiologic. The effect on H2AX are seen in the 1 (minimal) and 5 mM range (Figure 3) that you will not see it in any normal or pathologic condition.

AUTHOR RESPONSE: We have repeated the aldehyde challenge experiments in Figure 3 and Supplementary Figure 4 at lower concentrations of formaldehyde and acetaldehyde (see Figure 3c, 3e and Supplementary Figure 4c, 4d). We have added the following lines of description, rationale and references to the Methods (Page 22, line 498 - 503):

“Concentrations for the formaldehyde and acetaldehyde challenge experiments were selected based on literature evaluation of measured concentrations^{43,44}, and were within the measured concentrations from the tissue LC-MS data in the present study. To model the continuous production of endogenous aldehydes seen *in situ*, media containing fresh dilutions of the corresponding aldehydes was refreshed twice during incubation. “

By way of explanation, we chose the original concentrations based on recent publications investigating formaldehyde and acetaldehyde stress *in vitro* (Tan et al, Cell, 2017 PMID: 28575672; Zhang et al, Blood, 2016 PMID: 5159699). It is important to mention that the oesophagus is exposed to undigested exogenous aldehydes, which may be much higher than experienced elsewhere in the body in health and disease. However, we accept that modelling with the concentrations closer to those we observed in the present UPLC-MS/MS tissue is more convincing. Therefore, we have repeated the metformin-DNA damage response experiments using 150 uM formaldehyde and 500 uM acetaldehyde, compared to 300uM and 20,000uM respectively in the original submission. We were able to demonstrate the same effects, by using repeated exposures at these lower concentrations, in reduced serum conditions (see Figure 3 and Supplementary Figure 4).

REVIEWER: Formaldehyde is metabolized by an alcohol dehydrogenase.

AUTHOR RESPONSE: We believe that formaldehyde was enriched in EAC tissue because of the wider effects on redox, depletion of co-factors, and increased production, as we found no difference in the two formaldehyde metabolisers ALDH1L1 and ADH5 *in silico* (refer to Supplementary Figure 1b). However, fully elucidating this was beyond the scope of this work. We seek to address this complex mechanistic problem in the next phase of the work, having recently determined promising candidate source compounds (Abbassi-Ghadi et al, Cancer Research, 2020 PMID: 32345674). We have amended the Discussion accordingly (Page 18, lines 423-426):

“It will also provide a platform to explore how potential influences such as source compound concentrations, mechanisms of production, and microenvironmental factors contribute to the overall aldehyde phenotype, in addition to detoxification and repair.”

REVIEWER: The authors argument that their data are consistent with epidemiological studies that linked inactivating ALDH variants to esophageal malignancies, is rather weak and not aligned with their data. The references provided are for the ALDH2 and ALDH1A2 genes. No differences in ALDH2 expression are noted in their data.

AUTHOR RESPONSE: We have changed the text in the introduction and provided more appropriate references supporting our initial contention that esophageal glandular cells (Barrett’s, EAC) are poorly adapted to coping with aldehyde stress (Peng et al, Gut 2009, PMID: 18664505; Peng et al, Gut, 2012, PMID: 22157330, Page 4, Line 83-86):

“For esophageal squamous cancers, inactivating variants in ALDH2 convey predisposing risk¹⁰. For esophageal adenocarcinoma and the precursor esophageal intestinal metaplasia (“Barrett’s esophagus”), glutathione defenses – an auxiliary mechanism of aldehyde detoxification - may be depleted^{11,12}.”

By way of explanation, the original references were added to build the argument that esophageal tissue is required to cope with aldehyde stress. The link between inactivating *ALDH2* variants and esophageal squamous cancer is well established and contributes to this argument, so we kept the reference to the data. On reflection, we accept that the

ALDH1A2 data in EAC is not validated (either in this study or externally) and we have removed this reference.

As the reviewer states, we found no differences in *ALDH2* expression in esophageal tissue samples in this study, nor any other dataset we assessed. *ALDH2* variants have a role in the development of ESCC. We have not claimed that *ALDH2* has a role in EAC, which is a different disease, arising in the same organ. We have revised Discussion to clarify this (Page 17, Line 389-391):

“This may explain why *ALDH2* was not significantly different in this study, despite underactive *ALDH2* being associated with ESCC, a different type of esophageal malignancy¹⁰.”

REVIEWER: Based on the published data, there is no indication that patients with Sjogren-Larsson syndrome, caused by homozygous or compound heterozygous mutation in the *ALDH3A2* gene which encodes fatty aldehyde dehydrogenase, have any incidence of esophageal cancers.

We have added a line of explanation in the Results (Page 15, Line 324-327):

“Given that *TP53* inactivation occurs during progression to high grade dysplasia or EAC²⁷, this could explain why *ALDH3A2* loss was linked to progression in the immunophenotyping experiments (Supplementary Tables 4 and 6), although SLS observational studies have not reported increased EAC incidence²¹.”

In the present study we argue that *ALDH3A2* deregulation is a passenger effect to an extensively validated EAC genetic driver event (*TP53* deletion). Thus, *ALDH3A2* loss is associated with progression, but is not independently oncogenic. The main reason for highlighting SLS is that it demonstrates the non-redundant nature of *ALDH3A2*, meaning that changes in *ALDH3A2* expression can generate a metabolic phenotype. Refer to (Page 13, Line 292):

“Congenital *ALDH3A2* mutations cause Sjogren-Larsson syndrome (SLS), a neuro-cutaneous syndrome in which keratinocyte hyperplasia and fatty alkanal accumulation are features^{21,24}. This strengthens the argument that *ALDH3A2* is metabolically non-redundant.”

REVIEWER: Metformin, a listed as direct aldehyde scavenger but the paper cited (#11) mention a possible interaction between aldehydes and metformin.

RESPONSE: We have modified the abstract to state Metformin as a possible aldehyde scavenger (Page 3, line 51):

“Metformin, a putative aldehyde scavenger, reduced this toxicity.”

We have also provided the correct reference for direct interactions of metformin with acetaldehyde (Page 10, line 225, Ref 22: Deng, 2010, Luminescence, PMID: 21374792).

REVIEWER: • In Figure 2, the formaldehyde, 2-butanol and acetaldehyde data are not in agreement 4-HNE (genotoxic) data with the other data from medium chain aldehydes.

RESPONSE: In the Results we have described the different patterns of aldehyde concentrations, which differ between formaldehyde, acetaldehyde, and 1-butenal, versus HNE and the other medium aldehydes (refer to Page 8, Lines 192 - 196):

“The genotoxic aldehydes formaldehyde, acetaldehyde, and 1-butenal were significantly increased in both SqT and EAC compared to SqN (Figure 2c-e), suggesting aldehyde stress exhibits a field effect in the transformed esophagus. EAC was also enriched for other genotoxic aldehydes including glyoxal, malondialdehyde and 4-hydroxy-2-nonenal (HNE)”

REVIEWER: In most of the cases, high expression of ALDHs, especially *ALDH1A1*, *ALDH3A1* and *ALDH1B1* is associated with various cancers and genetic abolishment of these genes is associated with decreased cancer formation (e.g., PMID: 28881356, 26566217; 29767973, 28280079, 29767973). The authors should comment this as their results are the opposite

RESPONSE: We have added a new paragraph to the Discussion to provide appropriate comment, including the references mentioned by the Reviewer (Page 17, lines 383 – 401). This also includes two responses to two previous comments from the Reviewer.

“A recent pan-cancer analysis has found that expression of the ALDH isoenzymes differs greatly between cancers³⁶. A number of context-specific factors may mediate the net phenotype from deregulation of an individual ALDH gene, including (i) the activity of the other ALDH genes, particularly those with overlapping specificities, (ii) the redox state (iii) microenvironmental factors, including hypoxia and co-factor availability (iv) mechanisms of endogenous aldehyde production (v) the activity of collateral detoxification and repair mechanisms, for example the DNA damage response. This may explain why ALDH2 was not significantly different in this study, despite underactive ALDH2 being associated with a different esophageal malignancy¹⁰. In addition, increased expression of ALDH1A1, 1B1, 3A1 and others has been identified in other malignancies, with inverse associations with survival^{14,37,38}. The oncogenic function has been attributed to stemness (1A1) or chemotherapy resistance (3A1), and whether these changes are causal or reactive is unclear. In this study, ALDH1A3, -3A1, -4A1, and 9A1 were under-expressed in both Barrett’s epithelium and EAC, and the genotoxic aldehydes they detoxify were enriched. This suggests these phenotypes contribute to oncogenesis, and is in keeping with sequencing evidence of high mutational burden in both Barrett’s epithelium and EAC^{27,39}. In contrast, ALDH3A2 expression-loss was seen in EAC only, possibly as a result of tumor biology (i.e. TP53 copy loss), rather than the cause.”

Overall, it is a highly sophisticated paper in terms of methodology, approaches and clinical/translational significance. The authors should improve the biology of the paper especially in the discussion.

Vasilis Vasiliou

RESPONSE: We are very grateful to the reviewer for their insightful comments that have helped us further improve the manuscript. The Discussion is now extensively revised, especially with respect to the biology.

Reviewer #2 (Remarks to the Author):

I have a few comments, which I think the authors should address.

REVIEWER: to me the most important related to details of the pathological tumour staging of the EAC series from which the obtained breath sampled for breath analysis. Hence In other words 8th edition AJCC/UICC staging of EACs (esophagogastric junction cancers), the authors should indicate to the non-clinical scientific reads the spread in terms of the pathological staging of the tumours in their data base outlined simply below:

Tis High-grade dysplasia, defined as malignant cells confined by the basement membrane

T1 Tumor invades the lamina propria, muscularis mucosae, or submucosa

T1a* Tumor invades the lamina propria or muscularis mucosae

T1b* Tumor invades the submucosa

T2 Tumor invades the muscularis propria

T3 Tumor invades adventitia

T4 Tumor invades adjacent structures

The value of the study would depend on the percentage of T2, T3 and T4 as opposed to Tis, T1, T1a. T1b

RESPONSE: We have provided clinical metadata for the UPLC-MS/MS cohort and immunohistochemistry cohort, using 8th edition AJCC/UICC classification (refer to Supplementary Table 4 and 5).

REVIEWER: The study data showed that decanal is associated with more advanced tumour stage of EAC. Is this mirrored in the previous breath analysis study?

RESPONSE: The previous breath analysis papers did not provide subgroup analysis by stage (Kumar et al, *Annals of Surgery*, 2013 PMID: 25575255; Markar et al, *JAMA Onc*, 2019 PMID: 29799976). We provided a subgroup analysis of this data in Supplementary Figure 6b.

REVIEWER: Availability of underpinning technology (ultra-performance liquid chromatography tandem mass spectrometry) for measuring expired aldehydes. I am certain that the technology is unlikely to be available outside the large University Medical Schools, e.g., GP practices and district general hospital. Hence would there be an NHS reference laboratory for breath analysis of aldehydes and other volatile compounds?

RESPONSE: Our current methodology for breath collection uses thermal desorption tubes (Tenax/Carbograph-5TD sorbent phase, Markes International Ltd, Llantrisant, UK) with dimensions: 3½-inch (89 mm) long x ¼-inch (6.4 mm) diameter. This allows the transfer of a stable breath sample to a reference laboratory for analysis, with a stability of >two months. We have successfully deployed this technology in a multicentre study of 2500 patients (COBRA study; NCT03699163, data submitted for publication). Breath samples from those studies were transferred to a University Hospital for analysis. Our vision for future clinical practice is a similar hub-and-spoke model with samples analysed by a central accredited laboratory.

Reviewer #3 (Remarks to the Author):

The manuscript titled "Endogenous aldehyde accumulation generates genotoxicity and exhaled biomarkers in esophageal adenocarcinoma" focuses on describing and characterizing aldehyde biochemistry in the esophagus and how this may serve as a non-invasive tool for detecting esophageal adenocarcinoma (EAC). The manuscript is generally well written and the research topic is of strong interest to the scientific community. The research builds logically and incrementally on earlier work by the research team linking breath aldehyde to tissue and conducting limited *in vitro* work utilizing human EAC cell lines. However, portions of the data and results lack clarity, are not convincing and suffer from reduced scientific rigor. Thus, despite some interesting positive preliminary data, some concerns require resolution for the research findings to be fully appreciated.

Specific areas for improvement:

1. Abstract, it is premature to claim the results support aldehyde metabolism as a chemoprevention target given the results to date have been largely in late stage EACs and discerning between Barrett's progressors and non-progressors for example or BAR and EAC remains unclear. This comment applies to the discussion as well.

RESPONSE: We have revised the text in the Abstract and Discussion (concluding remarks) accordingly:

(Abstract: Page 3, lines 56-57): "These results support EAC early diagnosis trials using exhaled aldehyde analysis." (Removing previous text that aldehydes are a chemoprevention target)

(Discussion, concluding remarks: Page 19, lines 435-436): "In addition, these data also support the continued clinical investigation of exhaled aldehydes for EAC early diagnosis." (Removing previous text that aldehydes are a chemoprevention target)

In addition, we have strengthened *in vitro* data in which the protective effect of metformin against aldehyde stress is confirmed, both in BAR and EAC cell line models (see Figure 3c and 3d, and Supplementary Figure 4c, 4d, 4e and 4f).

2. Intro, line 74, ref 4 is from 2008, may consider updating the reference as it seems more current data supports that the increase in EAC incidence has leveled off in recent years. Also, pertains to 348. Alternatively add context.

RESPONSE: We have revised the Introduction, and provided the correct reference (Page 4, Line 73-74):

“especially as the EAC incidence in Western countries *has risen sharply in recent decades, and is not projected to fall*.”

We have also changed the text in the Discussion (Page 16, Line 352):

“This is of critical importance, as the incidence of EAC has increased sharply in recent decades....”

3. The justification for the choice of aldehyde related markers chosen for evaluation is summarized in Supp Figure 1. Specifically, Fig 1b is the basis for discerning differences across publicly available data sets linked to pathology. The authors identified ALDH1A3,-3A1,-3A2,-4A1 and -9A1 as under-expressed in EAC or glandular tissue compared to esophageal squamous mucosa. “Normal” should be added to ... eso sq epithelium, line 107 and it should be made clear that based on supp Fig 1b both Barrett’s and EAC under-express ALDH genes compared to normal eso.

RESPONSE: We have added “normal” before esophageal squamous epithelium as requested.

We have made it clear that both BAR and EAC under-express most ALDH genes compared to normal squamous, as indicated in Supp Fig 1b (Page 6, Line 117-119):

“Further assessments suggested that *ALDH3A2* was expressed in BAR but not EAC, whereas other ALDH genes were under-expressed in both BAR and EAC (Supplementary Figure 1b and 1c).”

4. Supp Figures 1d and 1e appear to have the very same/duplicated loading control bands which is not scientifically possible. Tubulin runs between 50 and 55 kDa making it nearly impossible to be the real loading control for some of the proteins assessed which

run at similar kDa. Representative images of loading controls are understood, but duplication is not acceptable.

RESPONSE: We thank the reviewer for pointing out this error that happened as a result of 1d & e being part of the same experimental run.

For Supplementary Figure 1e, we have repeated the experiments and subsequently revised Supp Figure 1e, including the correct loading control for ALDH isoenzymes, as well as additional Barrett's cell lines as requested elsewhere.

5. Scale/Enlarge figure 1C so it is legible upon printing.

RESPONSE: We have increased the font size of Figure 1c so it is legible on printing.

6. Supp 1d and 1e compares EAC cell lines to keratinocyte cell lines, but not normal esophageal cell lines or Barrett's cell lines which would be informative and more relevant. Moreover, Barrett's cell lines are readily commercially available.

RESPONSE: We have repeated the immunoblots and then revised Supplementary Figure 1e, including ALDH phenotyping data from three Barrett's cell lines as requested. We have not added the Barrett's lines to Supplementary Figure 1d however, as this experiment's purpose was solely to demonstrate that the keratinocyte cultures expressed expected squamous markers, and the EAC lines served as an appropriate control for this experiment. We have added information regarding the new Barrett's cell lines to Supplementary Table 1 (Materials).

7. Supp 1C, study groups appear normalized to a loading control-GAPDH, thus, what SqT/EAC notation indicate relative to the plot shown in supp Fig 1C is unclear? Indicate the statistical test utilized for significance in Supp Fig 1C, as there is great variation in data; yet, highly significant p-values.

RESPONSE:

(i) We have removed the confusing SqT/EAC notation, and revised the figure legend to make this point more clearly: "c. ALDH3A2 expression in the SqN and BAR tissues, compared to SqT and EAC data from Figure 1c."

(ii) We have revised the Supplementary Figure legend to detail the statistical test used:

“...Mann-Whitney U test...”

8. Description of ALDH3A results is confusing, line 119. The data that this discussion links to is unclear, please clarify.

RESPONSE: We have revised the Results to improve clarity (Page 7, lines 121 - 124):

“*ALDH3A1* was strongly expressed in SqT but weakly in Barrett’s tissue and EAC (Figures 1d and f). In contrast, its 17p co-localising homolog *ALDH3A2* was expressed in all SqT, >90% of Barrett’s metaplasia and dysplasia, and 21% of EAC (Figures 1e and f).”

9. Supp Fig 2 appears mislabeled as no Supp 2d is shown; yet, discussed in the text. Additionally, results referring to stabilization of HNE-protein adducts is not convincing. Sole use of OE33 cells may not reflect generalizable effects.

RESPONSE:

- (i) Labelling: We have added to Supplementary Figure 2 in response to another comment from the Reviewer, and ensured the labelling of all parts is correct in the Figure, Figure legend and text (see Supplementary Figure 2)**
- (ii) HNE-epitopes: We have deleted the HNE-epitopes data (see Supplementary Figure 2g), and deleted the reference to HNE-epitopes in the Results (Page 7, Line 169 - 170, “...and HNE-protein adducts..” is now deleted), and reference to the corresponding antibody in the Materials (see Supplementary Table 1).**
- (iii) Generalizability: We have added further in vitro data to demonstrate generalizability of the results:**
 - a. OE33 and FLO1 cells: The inhibitor, hypoxia, and inhibitor/hypoxia challenge experiments (see Supplementary Figure 2a-f)**
 - b. OE33 and CPD cells: ALDH interference experiments (FLO1 cells not used as they have trace ALDH expression (see Supplementary Figure 2g-i)**
 - c. OE33, FLO1, CPB, CPD cells: Metformin reduces formaldehyde and acetaldehyde-mediated H2AX phosphorylation (see Figure 3c,d)**
 - d. FLO1 and CPD cells: Metformin reduces acetaldehyde mediated induction of the DNA damage response (see Figure 3f), decreases acetaldehyde-mediated toxicity (see Supplementary Figure 4c,d), and decreases acetaldehyde-mediated depletion of glutathione (see Supplementary Figure 4e,f).**

10. Figure 2 results are unclear in that some of the bars are not marked as indicating significance or NS making it difficult to interpret the claims in the text.

RESPONSE: To improve clarity, we have removed all the “NS” notations so that only the significant differences are highlighted (see Figure 2 and Supplementary Figure 3).

11. Supp Fig 4, Proper controls appear missing for Fig 4C. Why are the levels of metformin to much higher in sup fig 4d compared to sup 4c—is this a relevant concentration and why is yet a different concentration of metformin used in supp 4e. The inconsistency makes interpretation difficult as dose surely has an influence.

RESPONSE: To aid interpretation, we have repeated all the metformin experiments in Figure 3 and Supplementary Figure 4, at the 0, 300 uM and 3mM concentrations of metformin.

- (i) We have repeated the viability experiment in another cell line and kept the concentrations of metformin consistent, including the 0uM control (see Supplementary Figure 4c,d)
- (ii) We have repeated the glutathione experiment at the correct metformin concentration (see Supplementary Figure 4e,f)

By way of explanation, the original Supplementary Figure 4d was performed at a range of concentrations of metformin and only the minimum concentration which showed an effect were provided. We fully accept that uniform experimental conditions aids interpretation, hence we have repeated the above experiments to achieve this.

12. Most of the cell culture work is done in a single cell line and not consistent across experiments (i.e. viability and drug challenge in Flo1, siRNAtargets in OE33) making it difficult to understand whether effects noted are limited to the selected specific cell line versus a panel of BAR or EAC cells.

RESPONSE:

We have repeated the following experiments in more than one cell line:

- a. DEAB, aldehydes: 2 lines (OE33, FLO1, see Supplementary Figure 2a,d)
- b. Hypoxia, aldehydes: 2 lines (OE33, FLO1, see Supplementary Figure 2b,e)
- c. DEAB + Hypoxia, aldehydes: 2 lines (OE33, FLO1, see Supplementary Figure 2c,f)

- d. siALDH, aldehydes: 2 lines (OE33, CPD, see Supplementary Figure 2g,h,i)
- e. Metformin chemoprotection: 4 lines (OE33, FLO1, CPB, CPD, see Figure 3c,d)
- f. Metformin reduces DNA damage response: 2 lines (FLO1, CPD, see Figure 3f)
- g. Metformin reduces acetaldehyde toxicity (proliferation): 2 lines (FLO1, CPD, see Supplementary Figure 4)
- h. Metformin reduces acetaldehyde toxicity (glutathione): 2 lines (FLO1, CPD, see Supplementary Figure 4)

The descriptive experiments in Supplementary Figure 2a-i merely demonstrate the point that loss of detoxification and hypoxia are sufficient to cause aldehyde accumulation, and thus can really be done in any esophageal model. The metformin experiments however needed to be disease-specific, and the anti-aldehyde properties of metformin have now been demonstrated in a panel of four BAR and EAC cell lines.

13. Fig 3 e, Western blot of poor quality/resolution and is not convincing. The blot should be repeated.

RESPONSE: The repeated experiment and revised Figure 3f is provided, with a further repeat in an additional cell line.

14. Despite the emphasis on decanal and claims made that decanal accumulation is affiliated with a metabolic phenotype there is no discussion of the rather opposing results showing that decanal does not differ in patients treated with metformin.

RESPONSE: Our claim that metformin may protect against reactive aldehydes (e.g. formaldehyde, acetaldehyde) through direct scavenging, and the data in Figure 3 and Supplementary Figure 4, with associated references, supports this. Decanal like other medium chain alkanals is relatively unreactive (which adds to its suitability as an exhaled biomarker), and the lack of difference in the diabetic cohort is unsurprising. We have not suggested that decanal is genotoxic or might react with metformin. We have amended the Discussion with following line (Page 18, Lines 413 - 415):

“The lack of difference in decanal concentration in diabetics taking metformin is unsurprising as decanal is relatively unreactive, which further supports its potential as a volatile biomarker.”

15. The claim that decanal can identify patients with treatable disease is overreaching.

RESPONSE: We agree, as it is one of five compounds in the best current biomarker model which can identify patients with treatable disease (JAMA Onc, 2019 PMID: 29799976). Therefore we have changed the Discussion accordingly (Page 18 , Lines 408 - 409):

“... and the breath test (five biomarkers, including decanal) can detect treatable disease stages.”

16. Correct, Figure 5 is out of order, two 5g figures are shown.

RESPONSE: We have corrected the relevant labelling error (see Figure 5). The labelling error was only in the Figure; thus, the Figure, Figure Legend and Results text are all now consistent.

REVIEWERS' COMMENTS

Reviewer #1 (Remarks to the Author):

The authors have addressed all of my concerns and comments satisfactorily. I therefore recommend acceptance of this manuscript.

Reviewer #3 (Remarks to the Author):

Thank you to the authors for thoughtfully addressing the initial reviewer concerns point by point resulting in a clarified much improved manuscript. The research reveals new information characterizing aldehyde biochemistry in the esophagus which holds potential as a noninvasive tool for detecting esophageal adenocarcinoma (EAC). Nice work!